# A dynamically based method for estimating the Atlantic meridional overturning circulation at 26°N from satellite altimetry

Alejandra Sanchez-Franks[1], Eleanor Frajka-Williams[1], Ben I. Moat[1], and David A. Smeed[1]

[1]National Oceanography Centre, University of Southampton Waterfront Campus, European Way, Southampton, SO14 3ZH, UK.

*Correspondence to*: A. Sanchez-Franks (alsf@noc.ac.uk)

**Abstract.** The large-scale system of ocean currents that transport warm waters in the upper 1000 m northward and return deeper cooler waters southward is known as the Atlantic meridional overturning circulation (AMOC). Variations in the AMOC have significant repercussions for the climate system, hence there is a need for long-term monitoring of AMOC fluctuations. Currently the longest record of continuous directly measured AMOC changes is from the RAPID-MOCHA-WBTS programme, initiated in 2004. The RAPID programme, and other mooring programmes, have revolutionised our understanding of large-scale circulation, however, by design they are constrained to measurements at a single latitude and cannot tell us anything pre-2004.

Near-global coverage of surface ocean data from satellite altimetry is available since the launch of TOPEX/Poseidon satellite in 1992 and has been shown to provide reliable estimates of surface ocean transports on interannual time scales including previous studies that have investigated empirical correlations between sea surface height variability and the overturning circulation. Here we show a direct calculation of ocean circulation from satellite altimetry of the upper mid-ocean transport (UMO), the Gulf Stream transport through the Florida Straits (GS), and the AMOC using a dynamically based method that combines geostrophy with a time mean of the vertical structure of the flow from the 26°N RAPID moorings. The satellite-based transport captures 56%, 49%, and 69% of the UMO, GS, and AMOC transport variability, respectively, from the 26°N RAPID array on interannual (18-month) time scales. Further investigation into the vertical structure of the horizontal transport shows that the first baroclinic mode accounts for 83% of the interior geostrophic variability, and the combined barotropic and first baroclinic mode representation of dynamic height accounts for 98% of the variability. Finally, the methods developed here are used to reconstruct the UMO and the AMOC for the time period pre-dating RAPID, 1993 to 2003. The effective implementation of satellite-based method for monitoring the AMOC at 26°N lays down the starting point for monitoring large-scale circulation at all latitudes.

## 1 Introduction

The Atlantic meridional overturning circulation (AMOC) is the large-scale oceanic circulation comprised of currents that carry warm, shallow water northward and return cold deep water southward. Variations in the AMOC's strength have a significant impact on the Earth's climate system (Srokosz et al., 2012) as the AMOC's northward transport of

warm surface waters release heat on the order of 1PW over the North Atlantic (Trenberth & Caron, 2001), key in maintaining the relatively mild winter climate of northwest Europe (Hall & Bryden, 1982; Pohlmann et al., 2006). On decadal time scale, the AMOC is identified as the underlying driver of latitudinal shifts in the Gulf Stream path (Sanchez-Franks & Zhang, 2015), and changes in the AMOC are a key driver of the Atlantic Multidecadal Variability and associated climate variability (Zhang et al., 2019; Moat et al., 2019). On longer time scales, coupled climate models predict significant weakening of the AMOC which results in a reduced northward heat transport associated with increased greenhouse gas forcing (Caesar et al., 2018).

One of the existing programmes monitoring the AMOC is the RAPID-MOCHA-WBTS (RAPID – Meridional Overturning Circulation and Heatflux Array – Western Boundary Time Series;  hereafter RAPID) 26°N mooring array, which has been measuring the meridional volume and heat transport at 26°N since 2004 (McCarthy et al., 2015; B. I. Moat et al., 2020a; Rayner et al., 2011; Smeed et al., 2014; Moat et al., 2020b; Johns et al., 2011). The RAPID programme is an international campaign that uses an array of moored instruments at strategic locations in order to continuously estimate AMOC fluctuations (further details in McCarthy et al., (2015)). Key to the RAPID strategy is that the interior of the ocean is largely in geostrophic balance; placing moorings in the western and eastern endpoints of the Atlantic basin, as well as both flanks of the mid-Atlantic ridge, allows for fluctuations in the vertical density profiles to be simultaneously measured at the mooring locations. By taking the difference between the endpoint measurements, the basin-wide interior flow can be estimated (Cunningham et al., 2007; Kanzow et al., 2010; Rayner et al., 2011). The vertical density profiles estimated from the RAPID moorings is affected by long Rossby waves that start off as westward-traveling perturbations (Johnson & Marshall, 2002), altering the east-west density gradient driving AMOC variability on seasonal and subannual time scales (Hirschi et al., 2007).

Other international efforts to measure the AMOC via mooring array programmes include: SAMBA (South Atlantic MOC/South Atlantic MOC Basin-wide Array) at 34.5°S (Meinen et al., 2018; Kersale et al., 2020), TRACOS (Tropical Atlantic Circulation and Overturning) at 11°S (Hummels et al., 2015; Herrford et al., 2021), and OSNAP (Overturning in the Subpolar North Atlantic Program) in the subpolar North Atlantic (Lozier et al., 2019). RAPID and these other mooring array programmes have made step-change advancements in our understanding of the AMOC over the last 15 years (Frajka-Williams et al., 2019), however they are limited to a single line of latitude and alone cannot be used to infer upstream/downstream changes in the larger scale structure of the AMOC. Further, the large meridional distances between the arrays (e.g. 34.5°S, 11°S, 26°N, 58°N), combined with the relatively short length/lifespan of the programs (e.g. the oldest is RAPID, initiated in 2004), preclude our ability to understand AMOC connectivity, i.e. its meridional coherence, which is still poorly understood (e.g. Bingham et al., 2007).

Satellite data provide high spatial coverage of the global oceans. Satellite altimetry measures data of the surface topography of the ocean (i.e. sea surface height; SSH), which is an effective tool for measuring large-scale surface circulation and associated dynamics. In particular, westward propagating Rossby waves, or surface intensified flow which is largely of baroclinic structure, are captured by satellite observations (Chelton et al., 2007). Previous efforts

to use Earth-observing methods have shown capacity in estimating the large-scale ocean circulation. For example, EOF analysis of satellite sea level anomaly (SLA) suggested altimetry could be useful indicator for AMOC transports along 40-50°N on interannual time scales (Bingham & Hughes, 2009). Frajka-Williams, (2015) found that the relationship between satellite SLA and RAPID dynamic height was robust enough to be used to create a proxy for the upper mid-ocean transport, which when combined with Ekman transport (derived from reanalysis products) and Gulf Stream transport (from a submarine telephone cable) accounted for more than 90% of MOC variability on interannual time scales. The method developed in Frajka-Williams (2015) was purely statistical. The relationship between dynamic height from RAPID moorings and satellite SSH at 26°N was also found to be robust (accounting for up to 72% of variance at a station 500 km offshore) (Kanzow et al., 2009), however this relationship was found to deteriorate in proximity to the western boundary (Bryden et al., 2009; Clément et al., 2014).

One of the limitations of satellite altimetry is that it only provides information about the sea surface. The information captured by satellite altimetry at the surface is representative of variability associated with the first baroclinic mode (Hirschi et al., 2007). Because of the baroclinic nature of the surface intensified ocean, Rossby wave theory suggests that some MOC variability can be linked to satellite altimetry on subannual (Hirschi et al., 2007) and subannual to interannual (Hirschi et al., 2009) time scales. However, satellite altimetry cannot tell us anything about the vertical shape of the horizontal velocity which forms part of the AMOC structure. To get around this, previous studies have combined satellite altimetry data with vertical structure of the flow derived from in situ datasets in the North Atlantic at 26°N (e.g. Hirschi et al., 2007, 2009; Kanzow et al., 2009; Szuts et al., 2012). In particular, Hirschi et al. (2009) found that combining SSH with baroclinic structure, using a numerical model, yielded a robust estimate of eastern and western branches of the MOC, but results could not be replicated for reconstructing variability of the basin-wide MOC on subannual to interannual time scales (Hirschi et al., 2007; Hirschi et al., 2009). They found that even small errors in the western and eastern components could overwhelm MOC variability, and only highly accurate estimates of the western and eastern branches would yield a reliable reconstruction of the basin-wide MOC. In a study focused on the impact of eddy dynamics on basin-wide transport, Kanzow et al. (2009) tested whether geostrophic transport could be obtained by combining geostrophic surface flow from SSH with the dominant vertical mode of horizontal velocity; they found that the correlation between the difference in SSH at eastern and western endpoints of the basin had little skill in capturing the upper mid-ocean transport and suggested the reason for this was not enough satellite data close to the shelf as well as changes in the vertical structure of the flow moving toward the western boundary. Szuts et al. (2012) similarly found that SSH captured fluctuations in the centre of the basin but struggled to capture the variability closer to the boundaries, where the vertical structure of the flow was not as well described by the dominant baroclinic mode.

The length of the RAPID records has more than doubled since the aforementioned studies that combined geostrophy and vertical structure of horizontal velocity (e.g. Hirschi et al., 2009; Kanzow et al., 2009; Szuts et al., 2012), so the time is right to re-examine the issues (e.g. problems at the boundary, including contribution from vertical modes) and build on the methods developed in these previous studies. Thus, here we combine geostrophic principles with the

vertical structure of the horizontal flow to develop a satellite-based MOC transport on interannual time scales. In contrast to earlier efforts by Hirschi et al. (2009) and Kanzow et al. (2009), the longer records now available at RAPID enable us to test the methods for longer timescales (interannual, rather than sub-annual), and we find that the skill is generally higher at interannual timescales than at shorter timescales. Unlike Frajka-Williams (2015), this method now uses geostrophy (east minus west differences in sea level anomaly) rather than finding the point location where the sea level anomaly has the strongest correlation with the transport variability. This change in the method is a prerequisite to developing similar methods at other latitudes as it reduces (though does not entirely eliminate) the requirement for in situ data to 'train' the method. In addition, while Szuts et al. (2012) and Clement et al. (2014) found that multiple modes are required to explain the variance in the western boundary profiles, we show that on interannual timescales, the contributions from higher modes are reduced and the first mode explains a majority of the variance. In the following section (2), a brief overview of data and methods used here are presented. Then an evaluation of the satellite data and parameters from the RAPID moorings follows in section 3. Rossby wave theory and an in-depth analysis of horizontal velocity normal modes from RAPID mooring data is shown in section 4. Sections 5 and 6 investigate construction of the upper mid-ocean, the Gulf Stream, and the AMOC transports from satellite altimetry. Finally, summary and conclusions are given in section 7.

**2 Data and Methods**

**2.1 Satellite and mooring data**

For this study, SLA from the Copernicus Marine Environment Monitoring Service (CMEMS: http://marine.copernicus.eu; product ID: SEALEVEL_GLO_PHY_L4_REP_OBSERVATIONS_008_047) gridded multimission satellite altimetry is used to create a proxy for AMOC transport. The CMEMS gridded altimetry includes data from the following satellite missions: Envisat, Geosat Follow On, Jason-1, TOPEX/Poseidon Interleaved. The satellite product has a ¼° resolution in space and monthly in time, though the resolution is limited by the underlying track spacing between altimeter passes.

The SLA is compared to data from the RAPID 26°N mooring array. The RAPID array is designed to measure the AMOC at 26°N by using strategically placed moorings at 3 key regions (McCarthy et al., 2015; Rayner et al., 2011). The first is in the western boundary, the second at the mid-Atlantic Ridge, and the third along the eastern boundary (see inset in Fig. 1a). The RAPID moorings provide continuous dynamic height profiles from conductivity, temperature and depth using MicroCAT CTDs. Temperature and salinity from individual MicroCAT records are vertically interpolated using the climatological profiles of dT/dp and dS/dp to produce a regularly-gridded profile of temperature and salinity on a 20 dbar grid (Johns et al. 2005; McCarthy et al. 2005). Dynamic height ($\phi$) is then calculated from the gridded density profiles as:

$$\phi(\text{p}) = \int_{4820}^{p} \delta\,(p')\,dp', \tag{1}$$

where $\delta$ is the specific volume anomaly. The western boundary used in the RAPID AMOC calculation includes moorings WB2, WBH2 and WB3 (concatenated into a single profile and hereafter referred to as West). When comparing with the SLA, data from the RAPID moorings WB2 (26.5°N and 76.75°W) and WB3 (26.5°N and 76.5°W) and West are used for the western component, and an amalgamation of data from the eastern boundary moorings between 13.75°W-24.22°W and 23.7-27.9°N (concatenated into a single profile (Chidichimo et al., 2010; McCarthy et al. 2015) and hereafter referred to as EB) are used for the eastern component (section 3 and 4) (Fig. 1a). Further details on the instrumentation and observational strategy of the RAPID mooring array can be found in McCarthy et al. (2015).

RAPID MicroCATs record on an hourly basis. These data are then postprocessed with a 2-day low pass filter to remove tidal fluctuations. The resulting data is used to compute transport estimates which are gridded at 12-hour resolution. A 10-day low pass filter is applied to final transport estimates (Kanzow et al. 2007). RAPID data used here includes temperature, salinity, pressure, dynamic height, as well as estimates of the upper mid-ocean and AMOC transports from 2004 to 2018 (B. Moat et al., 2020b), which can be obtained here: http://www.rapid.ac.uk/rapidmoc/.

In this study, all 12-hour and daily data are monthly averaged, the annual seasonal climatology is removed and then smoothed with an 18-month filter over the available time periods, except where indicated otherwise. The filter is a Gaussian weighted moving average filter, where endpoints are computed such that when the number of points available is fewer than the window size, the window is resized/truncated and the average is taken from the elements available in the truncated window. Clément et al. (2014) found that Rossby waves and eddies are important contributors to geostrophic transport on time scales of 3-8 months and Kanzow et al. (2009) found that eddies do not affect AMOC on low frequency (interannual to decadal) time scales, hence a 18-month filter is deemed appropriate for removing the influence of eddies. Finally, the mean over 2004 to 2018, which is the overlapping time period between RAPID and the SLA, is removed from their respective time series. When computing correlations and regressions, the linear trend from 2004-2018 is removed from the respective time series, and the statistical significance is determined using the effective degrees of freedom calculated as the length of each time series over the integral time scale of decorrelation (Emery and Thomson, 2001).

## 2.2 RAPID transport estimates and SLA

RAPID defines the AMOC transport at 26°N as the sum of three components: the Gulf Stream transport ($T_{GS}$), the Ekman transport ($T_{EK}$), and the upper mid-ocean transport ($T_{UMO}$) (McCarthy et al., 2015; Kanzow et al., 2010; Kanzow et al., 2007; Johns et al., 2011; Rayner et al., 2011):

$$T_{MOC}(t) = T_{GS}(t) + T_{EK}(t) + T_{UMO}(t), \tag{2}$$

The first component, $T_{GS}$, has been measured by submarine telephone cables in the Florida Straits since 1982 (Baringer & Larsen, 2001; Meinen et al., 2010)(https://www.aoml.noaa.gov/phod/floridacurrent/index.php). The second component, $T_{EK}$, is derived from zonal wind stress from ERA5 reanalysis products (Hersbach et al., 2020). The third component, $T_{UMO}$, is the sum of the western boundary wedge transport ($T_{WBW}$), the hypsometric mass compensation ($T_{EXT}$), and the internal geostrophic transport ($T_{INT}$) over the top 1100 m. $T_{WBW}$ is the northward transport measured between the Abaco Island continental shelf and the WB2 mooring. $T_{EXT}$ functions to compensate $T_{UMO}$ so that the net meridional flow is zero, and $T_{INT}$ is the internal geostrophic [southward] transport.

Through geostrophic balance, the meridional mass transport is proportional to the integrated pressure difference between the eastern ($P_E$) and western ($P_W$) basin endpoints such that:

$$T(t) = \frac{1}{f\rho_0} \int_{z_{ref}}^{0} [P_E(z,t) - P_W(z,t)]dz, \tag{3}$$

where $f$ is the Coriolis parameter, $\rho_0$ is reference density, and $z_{ref}$ is the reference level at -4740 m. Pressure at $z = -h$ is then related to sea level displacement and the vertical profile of density as follows:

$$P = \rho_0 g\eta - \int_{-h}^{0} \rho g \, dz, \tag{4}$$

where g is gravitational acceleration and $\eta$ is satellite SLA.

Satellite altimetry measures the surface geostrophic velocity, but surface geostrophic velocity does not give us information about the vertical structure of the horizontal velocity, thus time-varying geostrophic flow is combined with the first mode of horizontal velocity to obtain $T_{UMO}$ (section 4 and 5). The $T_{GS}$ is estimated using satellite altimetry and linear regression (section 5). The $T_{MOC}$ is constructed by adding the satellite-derived $T_{GS}$ and $T_{UMO}$ with the $T_{EK}$ obtained from ERA5 wind stress as per Eq. (2) (sections 5 and 6). In the following sections, the relationship between satellite and vertical structure of the flow from the moorings is used to develop a new method for estimating AMOC transport.

**3 Evaluation of the satellite and mooring data**

The CMEMS absolute dynamic topography (ADT) is computed as the sum of SLA and mean dynamic topography (MDT). The MDT is a mean estimate of SSH above the geoid over the given reference period 1993 to 2013 (Rio et al., 2018; further details on this data product here: https://www.aviso.altimetry.fr/en/data/products/auxiliary-products/mdt.html). The ADT shows characteristics of the mean state North Atlantic with negative ADT marking the subpolar gyre in the north (>= 48°N; Fig. 1a) and positive ADT delimiting the subtropical gyre, largely occupying the region between 15°N and 40°N (<40°N; Fig. 1a). Key features are observed at 36°N, where positive ADT represents the Gulf Stream as it separates from the coast at Cape Hatteras (Fig. 1a) and flows north as a free-wheeling jet,

eventually feeding into the North Atlantic Current on its path northward. The negative ADT around the region of the Grand Banks (48°N) is representative of the equatorward flowing Labrador Current that supplies the Slope Sea just north of the Gulf Stream (e.g. Petrie and Drinkwater, 1993; Frantantoni and Pickart, 2007) influencing its position (Pena-Molino and Joyce, 2008; Sanchez-Franks et al., 2016). The Labrador Current is also part of the surface flowing limb of the AMOC. The apparent bipolar structure over the North Atlantic is similar to the EOF mode 1 of the North Atlantic, characterized by Zhang (2008) as the AMOC fingerprint. The standard deviation of the ADT shows most of the variability is contained within the region of the Gulf Stream after it separates from Cape Hatteras (Fig. 1b). This variability is due to the large latitudinal shifts in the Gulf Stream position following changes in the North Atlantic Oscillation (e.g. Sanchez-Franks et al., 2014; Sanchez-Franks et al., 2015; Perez-Hernandez and Joyce, 2014, Bisagni et al., 2017; Taylor and Stephens, 1998). Further details of AMOC are reviewed in Zhang et al., (2019).

In the region of the RAPID 26°N mooring array (Fig. 1a,b), the time-varying ADT at western and eastern points show substantially higher mean and fluctuations in the western boundary (0.73 m root mean square (RMS)) compared to the eastern boundary (0.21 m RMS) over the 2004 to 2018 period (Fig. 1c). Variability from the cross-basin pressure gradient is largely driven by the variability along the western boundary on lower frequency (periods longer than a year) timescales (Frajka-Williams, 2015).

### 3.1 Surface variability

The RAPID programme estimates geostrophic transport from the difference between the basinwide eastern and western endpoint pressure fluctuations (Eq. 3). Correspondingly, to construct a $T_{UMO}$ estimate from satellite altimetry, the time-varying east-west SLA difference ($\Delta\eta$) is compared at each gridpoint with the RAPID $T_{UMO}$.

Fig. 2a shows the correlation between RAPID $T_{UMO}$ and $\Delta\eta$ at each gridpoint, where the eastern point remains fixed at the first easternmost SLA gridpoint at each latitude and the western point shifts longitudinally in the westward direction. RAPID $T_{UMO}$ and $\eta$ have had the annual seasonal climatology removed and an 18-month Gaussian filter applied. In general correlations are highest in the western part of the basin in the 26-30°N latitudinal band, and in the east for roughly the same latitudes between 25-35°W (Fig. 2a). It is interesting to note that the correlations along the RAPID line are lower than those found at higher latitudes 27-30°N. This could be due to an improvement in the signal to noise ratio further north.

The correlation map illustrates the optimal choice of longitude and latitude coordinates for max correlation of $\Delta\eta$ with RAPID $T_{UMO}$, r = 0.74 (statistically significant at 95% level), where the eastern SLA gridpoint is located at 27.875°N and 13.125°W and the western SLA gridpoint is located at 27.875°N and 74.375°W (Fig 2a). The east and west dynamic height measurements from the moorings make up the interior geostrophic component of the RAPID $T_{UMO}$ (section 2.2); however, the $T_{UMO}$ also includes contribution from the Antilles Current. For the satellite data to account for as much of the upper mid ocean transport variability as possible, it is advantageous to use satellite endpoints as

shown in Fig. 2a to calculate $\Delta\eta$, which here appears to better reflects the changes in the meridional mass transport. This is consistent with other studies which have similarly found higher agreement between RAPID $T_{UMO}$ transport and satellite variability north of 26°N (e.g. Frajka-Williams, 2015). Further, sensitivity of the correlation between RAPID $T_{UMO}$ and $\Delta\eta$ to the choice of filter was tested for 6 month intervals at 6 month, 12 months and 24 months, in addition to the 18 month Gaussian filter (Fig. S1). The pattern of correlations between RAPID $T_{UMO}$ and $\Delta\eta$ across the subtropical North Atlantic remained generally consistent between the four different choices of filter (as described above for the 18 month filter), where correlations decreased (for eastern SLA gridpoint located at 27.875°N and 13.125°W and the western SLA gridpoint located at 27.875°N and 74.375°W) and were lowest at r = 0.48 (statistically significant at 95% level) using 6 month filtering, increasing to r = 0.67 using 12 month filtering, and r = 0.79 when using the 24-month filter, contrasted with r = 0.74 when using the 18-month filter.

### 3.2 Variability in the vertical

In order to determine how well the SLA can estimate the upper 1000 m circulation, it is useful to assess to what depth the variability from SLA captures sub-surface variability. Thus SLA is compared to the dynamic height (Eq. 1) at every depth from the surface to 1100 dbar from the RAPID moorings in the western and eastern boundary. Fig. 2b shows the correlation (r values) between SLA west and east gridpoints (as indicated in previous section) and dynamic height from the RAPID moorings: West, WB3 and EB. In this section only, the respective RAPID mooring dynamic height has been referenced at the surface to SLA (hereafter $\phi_\eta$) to get correlation value of 1 at the surface which decreases with depth to assess to what depth surface variability is coherent. In the western boundary, SLA at 27.875°N and 74.375°W is compared with both WB3 and West, and the correlation in the top 1100 dbar is found to be everywhere greater than r = 0.79 (r = 0.79; statistically significant at 95% level). SLA has a higher correlation with $\phi_\eta$ at mooring West, compared to WB3, maintaining a statistically significant correlation coefficient above r = 0.88 (statistically significant at 95% level) in the top 1100 dbar. Correlation between SLA and WB3 decreases more abruptly at around 300 dbar. In the eastern boundary, correlation between SLA at 27.875°N and 13.125°W and $\phi_\eta$ at EB is similarly high throughout the top 1100 dbar of the ocean, albeit weaker compared with the western boundary mooring, with correlations from r = 1 at the surface decreasing to r = 0.77 (statistically significant at 95% level) at a depth of 1100 dbar (Fig. 2b). These results suggest that the variability observed at the sea surface is a good measure and coherent with variability to at least a depth of 1100 dbar. This is in agreement with Clement et al. (2014) who showed that isopycnal displacements at the RAPID western mooring locations agree well with satellite data. Specifically, Clement et al. (2014) did a similar analysis using profiles of isopycnal displacements, instead of dynamic height, from the moorings and satellite. They found statistically significant correlation (r = 0.5 - 0.6) between WB3 and WB2 with SSHA, respectively, in the upper 1000 m over 2004 to 2011. They did not analyse moorings in the eastern boundary. Given that here we have referenced the dynamic height to SLA at the surface, as well as a longer time period and different filtering, the discrepancies between our studies are reasonable.

For completeness, the SLA is also compared to dynamic height anomaly thickness for the 0-1100 dbar layer (Fig. S2). The dynamic thickness is calculated from the difference in dynamic height (referenced to 4820 dbar) at 0 and 1100

dbar and gives a measure of the shear between those levels (independent from choice of reference level). In both the western and eastern basin, the dynamic height thickness anomaly has a standard deviation smaller (0.30 $m^2\,s^{-2}$ at WB3 and 0.07 $m^2\,s^{-2}$ at East) than the SLA multiplied by gravitational acceleration (0.70 $m^2\,s^{-2}$ in the west and 0.14 $m^2\,s^{-2}$ in the east). This result suggests that the baroclinic structure in the upper 1100 dbar is not overwhelming the SLA signal, and the SLA can be used for transport estimates in the upper 1100 dbar.

## 4 Rossby wave theory and horizontal velocity modes

### 4.1 Westward propagation

Wind stress and density fluctuations dominate AMOC variability on seasonal and sub-annual time scales (Hirschi et al., 2007). Wind-driven variability may also play a role on interannual time scales, for example during the winter of 2009/2010, anomalous wind-driven Ekman transport contributed to interannual AMOC fluctuations (Zhao and Johns, 2014; Evans et al., 2017). However, density variability in the upper 1000 m has also been identified as a leading driver of the AMOC on interannual time scales (Hirschi et al., 2007). This density variability is associated with isopycnal perturbations that travel westward as long Rossby waves (Hirschi et al., 2007; Hirschi et al., 2009). The Rossby waves impact the upper mid-ocean component of the AMOC transport through their effect on the east-west density structure of the basin (Hirschi et al., 2007; Cabanes et al., 2008; Hirschi et al., 2009).

The changes in the upper 1000 m of the density field from westward propagation are visible as a sea surface signal and have been shown to be captured by satellite altimetry. Altimeters are most likely to reflect the first baroclinic mode, and by association motion of the main thermocline, due to the nature of baroclinic modes which is surface intensified (Wunsch, 1997). In the subtropical North Atlantic, studies have used satellite altimetry to track westward propagating anomalies (e.g. Clement et al., 2014; Hirschi et al., 2007; Kanzow et al., 2009). A Hovmoller of SLA along the 26°N RAPID line shows how perturbations in the eastern boundary propagate westwards up to 79°W before they reach the Bahamas and abruptly diminish (Fig. 3). Westward propagation is visible at monthly resolution and even to some extent with 18-month smoothing applied to the data (Fig. 3b). Amplitudes in the western boundary are larger compared to the eastern boundary, and particularly prominent during 1996-97, 2003-04, 2006, and post 2015. Interannual variations are observed in the western basin (between 70° and 80°W) and the strong positive anomalies apparent during 2003-2004 and negative anomalies in 2006-2007 are consistent with Kanzow et al., 2009 (their Fig. 5). The speed of these westward propagating anomalies is typically similar to baroclinic Rossby wave phase speed (Gill, 1982; Hirschi et al., 2007; Killworth & Blundell, 2003).

### 4.2 Modal decomposition

Though satellite altimetry can measure surface geostrophic velocities, it cannot be used to infer the vertical structure of the flow. To understand the contribution of the flow's vertical structure to the meridional mass transport, and how that impacts satellite-derived transport, pressure modes derived from RAPID mooring data are examined.


The vertical structure of the flow is assessed using normal mode decomposition, assuming a flat bottomed and motionless ocean (Gill, 1982), using data from the RAPID moorings at the western (West, WB3) and eastern (EB) boundary. The normal mode decomposition for the given mode number n = 0,1,2…is determined from the Sturm-Liouville Eq.:


$$\frac{d^2 G_n(z)}{dz^2} + \frac{N^2(z)}{c_n^2} G_n(z) = 0 \, , \tag{5}$$

$$F_n(z) = \frac{dG_n(z)}{dz} \, , \tag{6}$$

Where $N(z)$ is the Brunt-Vaisala frequency and $c_n$ is the phase/modal speed of the waves. The boundary conditions
are defined as $G_n = 0$ at $z = 0, -H$.

The buoyancy frequency ($N$) is estimated from temperature and salinity profiles at moorings West, WB3, and EB, which are converted into density profiles, $\rho(z)$, and averaged over the 2004 to 2018 time period:

$$N(z) = \sqrt{\frac{-g}{\rho_0} \frac{\partial \rho(z)}{\partial z}} \, , \tag{7}$$

Normalisation is then computed to satisfy Kronecker delta, $\delta_{ij}$, (rendering the resulting normal modes dimensionless) such that:

$$\int_{-H}^{0} F_i F_j \, dz = H\delta_{ij} \, , \tag{8}$$

Where $H$ is 4820 dbar, the bottom reference depth used in the AMOC calculation.

The buoyancy frequency and the pressure modes, $F_n$ (as per Eq. 5 and 6), are estimated at West, WB3 and EB (Fig.
4). The stratification profiles are characteristic of buoyancy profiles in the North Atlantic (e.g. Szuts et al., 2012; Clement et al., 2014): in the west, the buoyancy frequency shows a sharp maximum at 80 dbar and a second maxima around 600 to 800 dbar before decreasing to background stratification levels around 1100 dbar. These two peaks indicate strong stratification linked to the seasonal and main pycnocline (Siegel et al., 1999). Moving eastward across the basin, at WB3, the upper stratification decreases and the peak deepens to 100 dbar, the second peak also deepens
and has slightly lower stratification than at mooring West. The stratification minima apparent in West and WB3 at 300-400 dbar is indicative of 18°C water (i.e. subtropical mode water), characteristic of the western subtropical North Atlantic. On the other side of the basin, in the eastern boundary, the stratification at EB is decreased though the subsurface maximum remains at 100 dbar.

Figure 4 also shows the first four pressure modes at West and WB3 and EB. Mode 0 is here representative of the barotropic mode, while modes 1 to 3 are the first 3 baroclinic modes. The first mode shows a zero crossing roughly between 1000 and 1100 dbar in the western boundary (West and WB3), which deepens to 1500 dbar by the time it reaches the eastern boundary (EB). In general, western moorings show shallower zero crossings and more complex structure, reflective of the peaks in stratification in the west, compared to the eastern mooring which has a deeper zero

crossing and smoother structure. The vertical structure of the horizontal velocity shown here is consistent with the characteristic shape of known modes in the western North Atlantic (at this latitude) (e.g. Gill, 1982; Szuts et al. 2012, their Fig. 6.14c and 2, respectively). The shape of the first baroclinic mode in the surface to its first zero crossing (1100-1500 dbar) will be key in informing the satellite-derived estimates of transport, as altimetry has been shown to capture the upper 1000 m intensified structure of the flow (away from the boundaries by ~50 km or more), reflective

of the first baroclinic normal mode (Wunsch 1997; Szuts et al., 2012). The zeroeth mode, i.e. the barotropic mode, is important in the deeper ocean (> 1000 m), which is less stratified (Wunsch, 1997; Kanzow et al., 2008). However, below the upper 1000 m, changes in the deeper transport are not captured by satellite altimetry (Kanzow et al., 2008).

In the following sections, in-depth analysis of the normal modes from moorings and their relationship with satellite

altimetry is explored.

**4.3 Modal amplitude and variability**

To understand the importance of each mode to the total variability observed in dynamic height anomaly, $\phi$ (Eq. 1,

referenced to 4820 dbar), measured by the RAPID moorings, the modal amplitude is analysed and used to construct time varying $\phi$ from the normal modes. The amplitudes of the modes, $a_n$, are estimated by integrating the product of the normal mode, $F_n$, and $\phi$ such that:

$$a_n(t) = \frac{1}{H} \int_{-H}^{0} F_n(z)\phi(z,t)dz, \tag{9}$$


where $n = 0,1,2,3...$is the mode number and $H$ is the reference pressure 4820 dbar. Because satellite altimetry has been shown to reflect fluctuations associated with the amplitude first baroclinic mode (Hirschi et al., 2009; Wunsch & Stammer, 1997), the amplitude for the first mode, $a_1$, can also be estimated from SLA, $\eta$, such that:

$$\hat{a}(t) = s\frac{g}{[F_1(z=0)]}\eta(t), \tag{10}$$

Where $F_1(z=0)$ is the surface value of the first mode, and $s$ is an empirically determined scale factor equal to 0.25.

Because of baroclinic shear, SLA alone does not determine transport in the upper 1100 m of the water column: if we

integrated the SLA over the upper 1100 m to obtain geostrophic transport, the SLA would overestimate the transport magnitude compared to dynamic height (e.g. S3). For this reason, it is necessary to combine the SLA with a function

that describes the shape of the geostrophic transport profile. We have used the first baroclinic mode for this function, however, a comparison between the dynamic height from the RAPID moorings and the SLA indicate further correction/scaling is needed. Thus a scale factor is determined empirically by examining the signal from $\phi$ at the surface (z = 0) at moorings West and EB against $\eta$.

Fig. 5a shows the RMS dynamic height and sea level anomaly ($\eta$ multiplied by gravitational acceleration) at two latitudes across the Atlantic. At the eastern boundary, the RMS of g$\eta$ is about twice (0.18 m$^2$ s$^{-2}$) that of the dynamic height from the moorings (0.08 m$^2$ s$^{-2}$) (Fig. 5a). Moving west, RMS values of g$\eta$ peak at 74°W and 75°W (RMS of g$\eta$(27.875°N, 74°W) = 0.70 m$^2$ s$^{-2}$ and RMS of g$\eta$(26.125°N, 75°W) = 0.59 m$^2$ s$^{-2}$) before decreasing to the western boundary (RMS of g$\eta$(27.875°N, 77°W) = 0.56 m$^2$ s$^{-2}$ and RMS of g$\eta$(26.125°N, 77°W) = 0.39 m$^2$ s$^{-2}$). The rapid decrease in sea level variability at the western boundary is due to mesoscale suppression associated with the continental slope (Kanzow et al., 2009). The moorings, in contrast, show lower variance in dynamic height at both the eastern and western boundaries (RMS of $\phi$ (26.5°N, 76.75°W) = 0.31 m$^2$ s$^{-2}$ and RMS of $\phi$ (East) = 0.08 m$^2$ s$^{-2}$). A scatterplot of the east-west difference in surface $\phi$ (i.e. $\Delta\phi = \phi_{East}(z = 0) - \phi_{West}(z = 0)$) and the equivalent from sea level anomaly ($g\Delta\eta =$ g($\eta$(27.875°N, 13.125) $- \eta$(27.875°N, 74.375))) shows general agreement between the two values. The slope of the regression line is ~0.25, and the intercept goes through the origin (Fig. 5b). Fig. 5 suggests that the satellite altimetry alone does not capture the same magnitude of signal observed by the moorings. One of the reasons for this discrepancy may be due to the proximity of the moorings (e.g. WB2) to land, where variability experiences changes due to coastal processes (Kanzow et al., 2009). Kanzow et al. (2009) suggest that near (within 100 km) of the western boundary, variability becomes influenced by a combination/mixture of barotropic-baroclinic flow over the slope, reduced eddy variability, and coastally trapped waves. The value of the slope of the regression line, i.e. 0.25, is thus used as the scale factor for the satellite where indicated (e.g. Eq. 10).

The modal amplitude, $\hat{a}$, computed from $\eta$ (Eq. 10) is compared with the modal amplitude, $a_n$, computed from the first 3 baroclinic normal modes (Eq. 9) at West and EB (Fig. 6). The amplitude of the first mode, $a_1$, and $\hat{a}$ have a correlation of r = 0.57 (significant at 90% level) at West and a correlation of r = 0.67 (significant at 90% level) at EB. In the east (west), $\hat{a}$ is constructed using $\eta$ at 27.875°N and 13.125°W (27.875°N and 74.375°W). Comparison of $a_1$ and $\hat{a}$ time series shows a deviation in 2005-2006, where $\hat{a}$ has a larger negative anomaly (0.15 m$^2$ s$^{-2}$) compared to $a_1$ (0.02 m$^2$ s$^{-2}$); this is likely linked to the collapse of the WB2 mooring from November 2005 to March 2006 period, where WB3 mooring was used to fill the gap in the WB2 and West data. In general, all three modes start off with the same sign (negative) and eventually change sign (positive) with the exception of mode $a_2$ which starts positive. The modes have been defined/set to positive at the surface. The higher modes, $a_2$ and $a_3$, have smaller amplitudes compared to $a_1$.

The modal amplitude (Eq. 9) can also be combined with the pressure modes (Eq. 6) to reconstruct $\phi$ as follows:

$$\phi_n^*(z,t) = a_n(t)F_n(z) \tag{11}$$

Fig. 7 shows the reconstructed $\phi$, i.e. $\phi^*$, using only the first baroclinic mode at West and EB as per Eq. (11) from the surface to the reference pressure 4820 dbar over the 2004 to 2018 period. The first baroclinic mode, $\phi_n^*$ for n = 1, at West shows a general undulating pattern of negative and positive anomalies in the upper 1000 dbar: the anomalies are negative for years 2007-2008, 2011, 2014, and positive during 2009, 2013, and from 2015 to 2017, similar to the surface anomalies observed in $\phi$ at West (Fig. 7 a,b). The difference between $\phi$ and $\phi^*$ at West suggests that although

the reconstruction generally captures intensified structure in the upper 1000 dbar, it underestimates the magnitude of $\phi$ and does not capture any of the signal below 1000 dbar (Fig. 7c). These differences may be due to the fact that the reconstruction uses only the first baroclinic mode, reflective of changes in the main thermocline, and thus has oversimplified structure in the upper layer of the ocean compared to the $\phi$ at West. Further, the baroclinic structure, in general, is more likely to reflect changes in the stratified upper ocean to roughly 1000 m, while in the deeper less

stratified ocean, barotropic motion is more important (Kanzow et al., 2008). In the eastern boundary, $\phi^*$ is reconstructed using the first baroclinic mode at EB (Fig. 7d), similar to pressure perturbation analysis as per Szuts et al., (2012). The reconstruction captures similar negative and positive anomalies in the upper 1000 dbar to those observed in $\phi$ at EB (Fig. 7e). The magnitude of the $\phi$ anomalies are generally substantially smaller at the eastern part of the basin compared to the west.


The contribution of the barotropic mode (mode 0) and higher baroclinic modes (modes 1 and 2) to $\phi^*$ is examined in Figure 8. The inclusion of barotropic mode and higher baroclinic modes in $\phi^*$ shows substantially improved patterns of the $\phi$ variability in the upper 1000 dbar, with reduced differences at every pressure level (Fig. 8) compared to $\phi^*$ using only mode 1 (Fig. 7). To assess whether the inclusion of the barotropic and higher baroclinic modes makes a

meaningful contribution to the upper ocean transport, transports from the $\phi^*$ reconstruction using the normal modes and the $\phi$ at West and EB are presented in the following section.

**4.4 Transport anomalies from West and EB moorings**

Transport anomalies at West and EB are estimated using $\phi$, $\phi^*$ and $\eta$ respectively, with the aim of understanding the contribution of the normal modes to $\phi$ variability and implications for satellite-based transport. Kanzow et al. (2010), Chidichimo et al. (2010) and Szuts et al. (2012) set the precedent for estimating transport from a single mooring (as opposed to a horizontal gradient) to separate the contribution of the western and eastern components of the basinwide geostrophic transport. Therefore, transport from $\phi$ at West or EB (as indicated) is defined as:


$$T'_\phi(t) = \frac{1}{f} \int_{-1100}^{0} \phi(z,t) \, dz, \tag{12}$$

The mode-reconstructed $\phi^*$ transport is calculated using the modes (Eq. 6) and modal amplitude (Eq. 9) such that:

$$T'_F(t) = \sum T'_{Fn}(t) = \sum \frac{1}{f} a_n(t) \int_{-1100}^{0} F_n(z) dz, \tag{13}$$

If the amplitude of the first mode is taken to correspond to $\eta$ (i.e. $\hat{a}$ ; Eq. 10), then the satellite-based transport can be constructed using the vertical structure from the first baroclinic mode $F_1$ in the following manner:

$$T'_\eta(t) = \frac{1}{f}\hat{a}(t)\int_{-1100}^{0} F_1(z)dz, \tag{14}$$

A comparison of the $T_\phi$ and the reconstructed $T_F$ transports shows that $T_F$ accounts for 83% of the variability of $T_\phi$ (r = 0.91, statistically significant at 95% level) at West and 90% of the variance of $T_\phi$ (r = 0.95, statistically significant at 99% level) at EB, where $T_F$ is reconstructed using only the first mode derived from the dynamic height profiles

(Fig. 9 a,d). The satellite-based transport, $T_\eta$, on the other hand shows lower correlation with $T_\phi$ at both West (r = 0.60, statistically significant at 95% level) and EB (r = 0.49, statistically significant at 95% level). Though $T_\eta$ captures the general patterns of variability observed in $T_\phi$, there are several differences apparent between $T_\eta$ and $T_\phi$ occurring around 2005/06, which is when WB2 collapsed and data from WB3 was used instead; and from 2004-2006 at EB. Additionally, the $T_\eta$ time series generally appears to underestimate magnitude of the $T_\phi$ at West during 2009-2010,

2012 and 2016-2017 and at EB during 2010-2012, and 2016 (Fig. 9 a,d). The results shown here are comparable to a satellite and RAPID moorings comparison featured in Szuts et al., (2012; their Fig. 7), where correlation between SSH and geopotential anomalies from WB3 and WB2 where approximately r = 0.5, 0.7, respectively, and r = 0.75 at EB1 and close to zero at EBH. In Szuts et al., (2012) a 10-day low pass filter was used prior to correlation. Contrasted with results here, which include a longer time period, 18-month smoothing, and differences in the location of the SSH data,

it is not surprising there are differences to results from Szuts et al., (2012). Szuts et al., (2012) further posited those differences between mooring and satellite are likely due to a) loss of small spatial and temporal variability that occurs with satellite products, and b) satellite includes surface-layer processes which moorings do not entirely capture.

Including the second baroclinic mode as well as the first in the reconstructed transport, $T_F$, shows some improvement

in the correlation with $T_\phi$ at West (r= 0.98, statistically significant at 99%) and slight decrease at EB ( r= 0.94, statistically significant at 99%), though the amplitude of $T_F$ still underestimates $T_\phi$ (Fig. 9 b,e). The addition of the barotropic mode to the first baroclinic mode in $T_F$ noticeably improves the amplitude of $T_F$ and correlation with $T_\phi$ at West (r = 0.99) and EB (r = 0.999)(Fig. 9 c,f).

The basinwide geostrophic transport can be constructed for $\phi$ and $\phi^*$ respectively, by integrating the east – west difference such that Eq. (12) and (13) become:

$$T'_{\Delta\phi}(t) = \frac{1}{f}\int_{-1100}^{0} \phi_E(z,t) - \phi_W(z,t)\,dz, \tag{15}$$

$$T'_{\Delta F}(t) = \frac{1}{f}[a_{nE}(t)\int_{-1100}^{0} F_{nE}(z)dz - a_{nW}(t)\int_{-1100}^{0} F_{nW}(z)dz], \tag{16}$$


Where subscripts E and W denote east (EB) and west (West), respectively. $T_{\Delta F}$ and $T_{\Delta \phi}$ have a correlation of r = 0.91 (significant at 99% level) when $T_{\Delta F}$ is constructed using only mode 1 (Fig. 9g). The correlation between $T_{\Delta F}$ and $T_{\Delta \phi}$ increases to r = 0.97, when $T_{\Delta F}$ is constructed using modes 1 and 2, and to r = 0.99 when $T_{\Delta F}$ is constructed using modes 0 and 1 (Fig. 9h,i). These results suggest that the time-averaged first baroclinic mode accounts for most of the interior geostrophic transport variability, and the combined barotropic and first baroclinic mode accounts for 98%. The barotropic mode is reflective of changes in the deeper less stratified ocean.

## 5 Construction of the GS, the UMO and AMOC transports

The Gulf Stream within the Florida Straits has a mean of 31.2 Sv and is balanced by the UMO and Ekman transports, mean of -18 and 3.74 Sv, respectively, to yield the total mean AMOC transport of 17 Sv (McCarthy et al., 2015). The Gulf Stream time series is based on a submarine telephone cable that has been recording data between the Bahamas and Florida at 27°N since 1982 (Baringer and Larsen, 2001). Principles of geostrophy have been previously used to provide alternative mechanisms for estimating the cable-based $T_{GS}$ using satellite altimetry (Volkov et al., 2020) and pressure gauges (Meinen et al., 2020).

Here, the east-west difference in $\eta$ ($\Delta \eta$) in the western end of the basin (i.e. west of 77°W) is compared with the submarine cable data (Baringer and Larsen, 2001) over the RAPID time period (i.e. 2004 to 2018). Maximum correlation between $T_{GS}$ from the cable data and the satellite $\Delta \eta$ (r = 0.70, statistically significant at 95% level) is found when using $\eta_E$ located at 27.625°N and 77.125°W, and $\eta_W$ located at 27.625°N and 80.125°W (Fig. S4). Similarly, Volkov et al., (2020) used along-track satellite altimetry to infer $T_{GS}$ measured at the level of the Florida Straits, where they found that satellite altimetry captures 56% of variability observed from the $T_{GS}$ estimated from submarine cable records for the 2006-2020 time periods, where the cable-based transport estimates were subsampled at 10-day intervals to coincide with the along-track satellite passes. We use gridded altimetry with a similar objective (to determine the Gulf Stream transport via the Florida Straits from satellite altimetry) but with a focus on the lower-frequency variability rather than an instantaneous comparison with cable measurements.

Here, linear regression is used against the cable-based $T_{GS}$ to obtain the regression coefficients (*a, b*) needed to produce a final estimate for satellite-based GS transport:

$$T_{GS}^*(t) = a\Delta \eta + b, \qquad (17)$$

Where a = 8.0087, b = 0.0026, the eastern and western SLA points that correspond to $\Delta \eta$ are located at 27.625°N and 77.125°W for $\eta_E$, and at 27.625°N and 80.125°W for $\eta_W$. The accuracy of $T_{GS}^*$ was determined using a Monte-Carlo technique, where 90% of the time series was randomly sampled 10,000 times, performing a linear regression to obtain an upper (mean value of 0.0614 Sv) and lower (mean value of -0.0614 Sv) bound for the 90% confidence intervals.

The satellite-derived $T_{GS}^*$ captures 49% of the variability of the in-situ $T_{GS}$ and generally underestimates the amplitude of variability (Fig. 10a,b).

The satellite-based, $T_{UMO}^*$, is constructed by combining the satellite-based mode amplitude, $\hat{a}$ (Eq.10) with the first baroclinic mode $F_1(z)$ (Eq. 6) as follows:


$$T_{UMO}^*(t) = \frac{1}{f}\left[\hat{a}_E(t)\int_{-1100}^{0}F_E(z)\,dz - \hat{a}_W(t)\int_{-1100}^{0}F_W(z)\,dz\right], \tag{18}$$

Where subscripts E and W denote east (for $\eta$ at 27.875°N and 13.125°W and $F_E$ at EB) and west (for $\eta$ is 27.875°N and 74.375°W and $F_W$ at West), respectively. The satellite-derived $T_{UMO}^*$ is compared with RAPID $T_{UMO}$ time series

(r =0.75, statistically significant at 95% level) in Fig. 10b. $T_{UMO}^*$ follows the weakening apparent in the RAPID $T_{UMO}$ in the year 2006 (positive anomalies indicate weakening in southward transport and viceversa), followed by the sharp intensification of the southward flow reaching its maximum in 2009-2010. These changes in the transport have been previously documented in Smeed et al. (2014, 2018), and in particular the 2008-2010 intensification in RAPID $T_{UMO}$ is linked to strengthening of southward circulation in the main thermocline (Smeed et al., 2014). In general, the satellite

$T_{UMO}^*$ follows the overall pattern of variability measured by the $T_{UMO}$, capturing the weakening of southward flow in 2005-2006, 2011, and intensification of southward flow in 2009-2010 and 2016-2017. However, the $T_{UMO}^*$ does not capture the full extent of the RAPID $T_{UMO}$ intensification in 2009-2010 or in 2012 by 1 Sv (Fig. 10 c,d). Some of the missing variability may be linked to the barotropic component not captured by satellite altimetry, as shown in the previous section (4.4). The 90% confidence interval for $T_{UMO}^*$ was estimated using a Monte-Carlo technique, where

90% of the $\Delta\eta$ timeseries is randomly sampled 10,000 times, performing the regression with $\Delta\phi$ to acquire the scale factor as outlined in section 3.3, giving the upper and lower bounds for our calculation of $T_{UMO}^*$. Sensitivity of these results to the choice of filtering window was tested. We found that the correlations decreased for $T_{GS}^*$ (r = 0.6, r = 0.68) $T_{UMO}^*$ (r = 0.42, r = 0.67) when using a 6-month Gaussian filter and a 12 month Gaussian filter, respectively (Figs. S5, S6), compared to the 18-month Gaussian filter. Conversely, correlations increased for $T_{GS}^*$ (r = 0.73) $T_{UMO}^*$ (r = 0.79)

when using a 24-month Gaussian filter (Fig. S7).

$T_{MOC}^*$ is the sum of southward flow $T_{UMO}^*$, the Ekman transport, and northward-flowing $T_{GS}$. By incorporating the satellite-derived $T_{GS}^*$ as per Eq. (17) and the satellite-derived $T_{UMO}^*$ as per Eq. (18), $T_{MOC}^*$ can be defined by re-writing Eq. (2) such that:


$$T_{MOC}^*(t) = T_{GS}^*(t) + T_{EK}(t) + T_{UMO}^*(t), \tag{19}$$

Where the Ekman transport ($T_{EK}$) is derived from ERA5 zonal wind stress as per the RAPID time series method. The satellite-constructed $T_{MOC}^*$ and the RAPID $T_{MOC}$ have a correlation of r = 0.83 (statistically significant at 95% level;

Fig. 10e). $T_{MOC}^*$ captures the start of the RAPID $T_{MOC}$ in 2004, the weakening in 2005, the subsequent intensification in 2006, followed by the gradual weakening of RAPID $T_{MOC}$ reaching a maximum low in 2010, and back to zero in

2011. In 2010 and 2013, the $T^*_{MOC}$ noticeably underestimates the magnitude (> 1 Sv) of the RAPID $T_{MOC}$ fluctuations. The anomalously low event in $T_{UMO}$ during 2009-2010 is attributed to anomalous wind-driven Ekman transport (Roberts et al., 2013; Zhao and Johns, 2014). After the 2009-2010 event, the AMOC recovers and appears to show a positive tendency. The gradual increase in the $T_{MOC}$ apparent from 2010 onwards, however, has not yet been shown to be statistically significant at 26°N (Moat et al., 2020b). Using a smaller filtering window, which would allow for higher frequency variability, we find the correlation for $T^*_{MOC}$ decreases (r = 0.71, r = 0.81) to for 6-month and 12-month Gaussian filters (Fig. S5, S6), respectively, but increases (r = 0.85) when using a 24 month Gaussian filter (Fig. S7).

The satellite-derived $T^*_{UMO}$ and $T^*_{MOC}$ were also computed with a time varying $F_1(z,t)$ with almost no improvement: the correlation between $T^*_{UMO}$, using a time-varying $F_1$, and the RAPID $T_{UMO}$ increased slightly from r = 0.75 to r = 0.76, and for $T^*_{MOC}$ and RAPID $T_{MOC}$ there was no change (i.e. r = 0.83). This suggests that 69% AMOC variability can be estimated from SLA and the time-averaged pressure modes on interannual time scales.

## 6 Construction of volume transports from historical altimetry data

TOPEX/Poseidon was one of the first satellite altimetry missions to be launched with a major oceanographic focus (Fu et al., 1994); since its launch in 1992, it has provided an invaluable source of measurements for studying surface ocean circulation (e.g. Frajka-Williams, 2015; Willis, 2010). In this section, the $T^*_{UMO}$ and the $T^*_{MOC}$ are constructed for the full available satellite time period: 1993 to 2018 (full annual coverage in the satellite data for the product used here starts in January 1993), yielding 11 years of data that pre-dates the RAPID mooring array.

Previously, satellite-derived estimates of the $T_{UMO}$ and the $T_{MOC}$ have been compared with the RAPID 26°N array, and constructed for the 1993 to 2013 period in Frajka-Williams (2015; hereafter EFW15). EFW15 found that SLA could be used to obtain a proxy for mass overturning transport. The statistical relationship between the RAPID $T_{UMO}$ and the SLA on low frequency (18 month) timescales provided the main method for estimating the EFW15 $T_{UMO}$. The EFW15 $T_{MOC}$ was estimated by adding the same estimates of $T_{EK}$ (ERA5 reanalysis) and $T_{GS}$ (submarine cable) used by RAPID to the $T_{UMO}$. Here it is fitting to compare the EFW15 method with the method derived in sections 4 and 5, which relies on altimetric data and geostrophic balance, and therefore gives a methodologically consistent (satellite only) method that incorporates climatological stratification and has the potential to be applied to other latitudes. This is in contrast with EFW15 which was a statistically based method that assumes the existence of in situ observations. Further work is needed to determine the origins of the scale factor required to balance the transports, whether due to sampling issues or uncertainties in satellite sea level anomaly near the boundaries. Additionally, the RAPID $T_{MOC}$ is compared with a third separate $T_{MOC}$ estimated from GloSea5, a global ocean and ice reanalysis product (Blockley et al., 2014; Jackson et al., 2016; MacLachlan et al., 2015; Jackson et al., 2019). GloSea5 uses the ocean model NEMO (Nucleus for European Modelling of the Ocean) that has a ¼° surface resolution

and assimilates in observational data including satellite (Megann et al., 2014). The GloSea5 $T_{MOC}$ estimates at 26°N are averaged monthly (further details on GloSea5 described in (Jackson et al., 2019)). As in previous sections, annual seasonal climatology is removed from each time series and an 18-month Gaussian filter is applied (with the exception of the EFW15 time series which uses an 18-month Tukey filter). Since the overlap between the RAPID array data and the EFW15 time series spans 2004-2013 only, all correlations presented in this section reflect the r-value for the 2004 to 2013 time period.

The $T^*_{UMO}$ and the EFW15 $T_{UMO}$ are first compared with the RAPID $T_{UMO}$ time series in Fig. 11a. Over the 2004 to 2013 time period, the $T^*_{UMO}$ and EFW15 $T_{UMO}$ have correlations of r = 0.79 and r = 0.85 (statistically significant at 90% level), respectively with the RAPID $T_{UMO}$ time series. Both time series capture the RAPID $T_{UMO}$ weakening in 2005-2006 (positive anomalies indicate weakening in southward transport and viceversa), 2010-2011 and the strengthening in 2009-2010 and 2013 (Fig. 11a). Both the $T^*_{UMO}$ and EFW15 $T_{UMO}$ underestimate the 2009-2010 intensification observed by the RAPID $T_{UMO}$. In the time period pre-dating the RAPID programme, EFW15 $T_{UMO}$ shows a weakening in 1995 and 2001, and intensification in 1999 and 2004 of the southward flow. In comparison, the $T^*_{UMO}$ shows lower amplitude variability but a higher southward transport in the 1993-2003 time period. The EFW15 $T_{UMO}$ is centered on a single SLA point in the western end of the basin at 70°W and 30°N, while in the satellite estimates derived here, $T^*_{UMO}$ is derived from the difference in SLA in the east at 13.125°W and 27.875°N and west 74.375°W and 27.875°N (i.e. variability in the eastern end of the basin is taken into account as well). Thus holding the eastern component of the $T^*_{UMO}$ constant highlights its contribution to total $T^*_{UMO}$ variability, which in the pre-RAPID time period acts to strengthen the mean $T^*_{UMO}$ southward transport; in other words, holding the eastern component constant leads to a weakening in southward transport that suggests improved agreement with the magnitude of EFW15 $T_{UMO}$ (Fig. 11a).

The $T^*_{MOC}$, EFW15 $T_{MOC}$, and GloSea5 $T_{MOC}$ are compared to the RAPID $T_{MOC}$ over the 2004-2013 time period (Fig. 11b). The $T^*_{MOC}$ has the highest correlation with the RAPID $T_{MOC}$ (r = 0.87, statistically significant at 95% level) while EFW15 $T_{MOC}$ and GloSea5 $T_{MOC}$ show slightly lower correlations with the RAPID $T_{MOC}$ (r = 0.84 and r = 0.85, respectively, statistically significant for at least 90% level). The GloSea5 $T_{MOC}$ proxy underestimates the $T_{MOC}$ mean values in the 2004 to 2010, and does not capture the weakening of southward flow apparent in the RAPID $T_{MOC}$ in 2005-2006 (Fig. 11b). In the pre-RAPID time period, 1993-2003, there is surprisingly little agreement between the $T_{MOC}$ proxies. The EFW15 $T_{MOC}$ reconstruction shows a weakening in the southward flow in ~1996-1997 (positive anomaly), while the $T^*_{MOC}$ shows strengthening in the southward flow in 1997 (negative anomaly). None of the reconstructions quite agree with the general mid-to-late 90s strengthening in the AMOC, associated with changes in the Atlantic Multidecadal Variability phase, marked by a positive phase in the late 1990s, discussed by Zhang et al. (2019). In 2003-2004, GloSea5 $T_{MOC}$ and $T^*_{MOC}$ show a weakening closer to the $T_{MOC}$ values at the start of the RAPID programme, which is not captured by the EFW15 $T_{MOC}$. In general, it is difficult to determine which of the three $T_{MOC}$ reconstructions might provide a best estimate of the observed $T_{MOC}$ over the 1993-2004 time period. For instance, the correlations between each $T_{MOC}$ reconstruction and RAPID $T_{MOC}$ are very similar, however there is poor agreement

between the three $T_{MOC}$ reconstructions in the time period pre-dating RAPID. Reasons for this disagreement may be due to overfitting. In the case of the satellite-based transport reconstruction, EFW $T_{MOC}$, the statistical method used to estimate transport relied on RAPID mooring data and resulting time series in the period pre-dating RAPID, 1993-2003, may be biased by overfitting of data during the RAPID period, 2004-2013. Similarly, $T_{MOC}^*$ may be biased by choice of the scale factor, here based on comparisons between satellite and moorings during the RAPID time period. Further investigation is therefore needed to confidently reconstruct the MOC for the 1993 to 2003 time period.

**7 Summary and conclusions**

Mooring array programmes such as the RAPID 26°N array have made step-change advancements in our understanding of the AMOC; however, they are limited to measurements at a single latitude and alone cannot be used to infer upstream/downstream changes in the larger scale structure of the AMOC. Mooring arrays are also a single point of failure should the moorings collapse. In light of the increase in studies linking the AMOC to climate impacts (Zhang et al., 2019), it is now more important than ever to find long term and cost-effective replacements/backup systems that can monitor changes in the AMOC.

One previous effort to use satellite altimetry as a proxy for AMOC variability, led by EFW15, relied on a statistical relationship between the RAPID $T_{UMO}$ and the SLA on low frequency (18 month) timescales to estimate the $T_{UMO}$. The EFW15 $T_{MOC}$ was estimated by adding the same estimates of $T_{EK}$ (ERA5 reanalysis) and $T_{GS}$ (submarine cable) used by RAPID to the $T_{UMO}$. Their method successfully recovered ~85% of the $T_{UMO}$ and 92% of the $T_{MOC}$ on interannual time scales. Earlier studies used more dynamically based methods and combined SLA with the vertical structure of horizontal flow from mooring data and found that the SLA had little skill at capturing the upper mid-ocean transport (e.g. Hirschi et al., 2009; Kanzow et al., 2009; Szuts et al., 2012). The goal of this study was to similarly use satellite altimetry as a proxy for AMOC variability and re-evaluate dynamics-based methods (e.g. Hirschi et al., 2009; Kanzow et al., 2009; Szuts et al., 2012) for using satellite altimetry to estimate the $T_{UMO}$ as well as the $T_{GS}$ components of the AMOC, which could eventually be used at other latitudes. Thus, using principles of geostrophy and normal mode decomposition a method for constructing the upper mid-ocean transport, $T_{UMO}^*$, and by extension the $T_{MOC}^*$, at 26°N from satellite altimetry on low frequency (18 months) time scales was devised here by combining the first baroclinic mode, derived from time-averaged density profiles from the RAPID moorings, with SLA to reproduce the $T_{UMO}^*$ and $T_{MOC}^*$ transports. Using this new method, we find that 56% of RAPID $T_{UMO}$ variability could be captured and 69% of the RAPID $T_{MOC}$ variability on interannual (18 month) time scales, significantly more than at the shorter time scales studied by Kanzow et al (2009). Sensitivity testing using different filtering windows decreases variance captured by $T_{UMO}^*$ and $T_{MOC}^*$, respectively, to 18% of $T_{UMO}$ and 50% of the $T_{MOC}$ when using 6 month Gaussian smoothing; 45% of $T_{MOC}$ and 66% of the $T_{MOC}$ when using 12 month Gaussian smoothing; and an increase to 62% of $T_{UMO}$ and 72% of the $T_{MOC}$ when using 24 month Gaussian smoothing. The $T_{GS}$ component of $T_{MOC}$ was also reproduced separately using the satellite altimetry; we found that it captures 49% of the GS variability via the Florida

Straits as measured from the telephone cable (Meinen et al., 2010) on interannual time scales, similar to a recent study by Volkov et al. (2020), who also used satellite to estimate GS variability in the Florida Straits. Finally, the satellite-derived $T^*_{UMO}$ and $T^*_{MOC}$ are reconstructed for the full satellite period (1993 to 2018). The 26-year reconstructions of $T^*_{UMO}$ and $T^*_{MOC}$ were compared with another separate satellite-derived proxy (EFW15) and a $T_{MOC}$ reconstruction from the GloSea5 reanalysis. Though all reconstructions had correlations of at least r = 0.79 with RAPID-based transports for the 2004 to 2013 time period, the 1993-2003 time period showed poor agreement between the three $T_{MOC}$ reconstructions, suggesting further studies are needed to confidently reproduce the fluctuations in the AMOC during times pre-dating the RAPID programme.

Because the altimetry has no knowledge of vertical shear, estimating transport over the upper 1000 m using only satellite altimetry results in an overestimation of the transport's magnitude (Fig. S3). Thus, normal mode decomposition was investigated using the RAPID moorings to answer 2 questions: a) How much does the vertical structure of the flow contribute to the upper-mid ocean transport ($T_{UMO}$) variability, and b) can the pressure modes be combined with the satellite to provide an improved way of estimating mass overturning transport. In the first instance, we find that the first baroclinic mode accounts for 83% of the observed interior geostrophic transport variability, and the combined barotropic and first baroclinic mode account for 98% of the total variability. In the second instance, we find that combining the satellite altimetry with the vertical structure (from the 1st baroclinic mode) improves the magnitude of values for the altimetry-based transport. However, a scale factor is still needed to further correct the values to capture the magnitude of the $T_{UMO}$. We posit that the need for a scale factor is because of the proximity of moorings to land. Close to the coast the SSH signal is not so well resolved by altimetry and is significantly affected by the barotropic mode as well as the baroclinic mode (Kanzow et al., 2009). It is also important to note that combining the altimetry with the vertical structure (from 1st baroclinic mode) does not improve the amount of variance captured by the satellite-based $T^*_{UMO}$ compared to RAPID $T_{UMO}$ variability, and using time-varying pressure modes (instead of time-averaged) do not necessarily improve correlation between the RAPID $T_{MOC}$ and the satellite-derived $T^*_{MOC}$. Further discrepancies between satellite-based transport and RAPID transport may be due to the barotropic component, which the satellite-based method does not account for.

To summarise, principles of geostrophy can be used with satellite altimetry to effectively capture upper mid-ocean and GS transport, and by extension, AMOC transport. The vertical structure of the flow (from the moorings) does not improve the amount of variability the satellite-based method captures; however, the analysis of the normal mode decomposition yields insight into the governing modes of variability associated with the $T_{UMO}$ on interannual timescales, and the limits this places on any satellite-based transport method. While this method progresses attempts at a dynamically robust method for estimating AMOC transport using satellite altimetry, which can also be used to recreate the AMOC the start of the satellite time period, pre-dating RAPID, the method is still not independent from the mooring array and without improved understanding of the scale factor, further investigation is needed to produce an altimetry-based method for the AMOC that can be used at other latitudes.

**Data availability**

The RAPID-MOCHA-WBTS time series (B. Moat et al., 2020) is available at https://doi.org/10.5285/aa57e879-4cca-28b6-e053-6c86abc02de5. Copernicus Marine Environment Monitoring Service for data access available online: http://marine.copernicus.eu/services-portfolio/access-to-products/?option=com_csw&view=details&product_id=SEALEVEL_GLO_PHY_L4_REP_OBSERVATIONS_008_047. ERA5 wind stress is available via https://cds.climate.copernicus.eu/cdsapp#!/home (Copernicus Climate Change Service, 2020). The Gulf Stream cable data is available via https://www.aoml.noaa.gov/phod/floridacurrent/index.php. The GloSea5 time series is available from Jackson et al., (2019) upon request.

**Author Contributions**

ASF wrote the manuscript with input/contributions from all authors. ASF, EFW, DAS, BIM contributed to the analysis. EFW, BIM, DAS contributed to data collection/provided processed data.

**Competing interests**

The authors declare that they have no conflict of interest.

**Acknowledgements**

The authors thank the reviewers for their helpful comments. The authors also thank the many officers, crews and technicians who helped to collect these data. ASF also thanks L. Clement for helpful discussions on normal mode decomposition.

**Financial support**

This research has been supported by grants from the UK Natural Environment Research Council for the RAPID-AMOC program, the ACSIS program (grant no. NE/N018044/1), and funding from the European Union Horizon 2020 research and innovation programme BLUE-ACTION (Grant 727852).

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

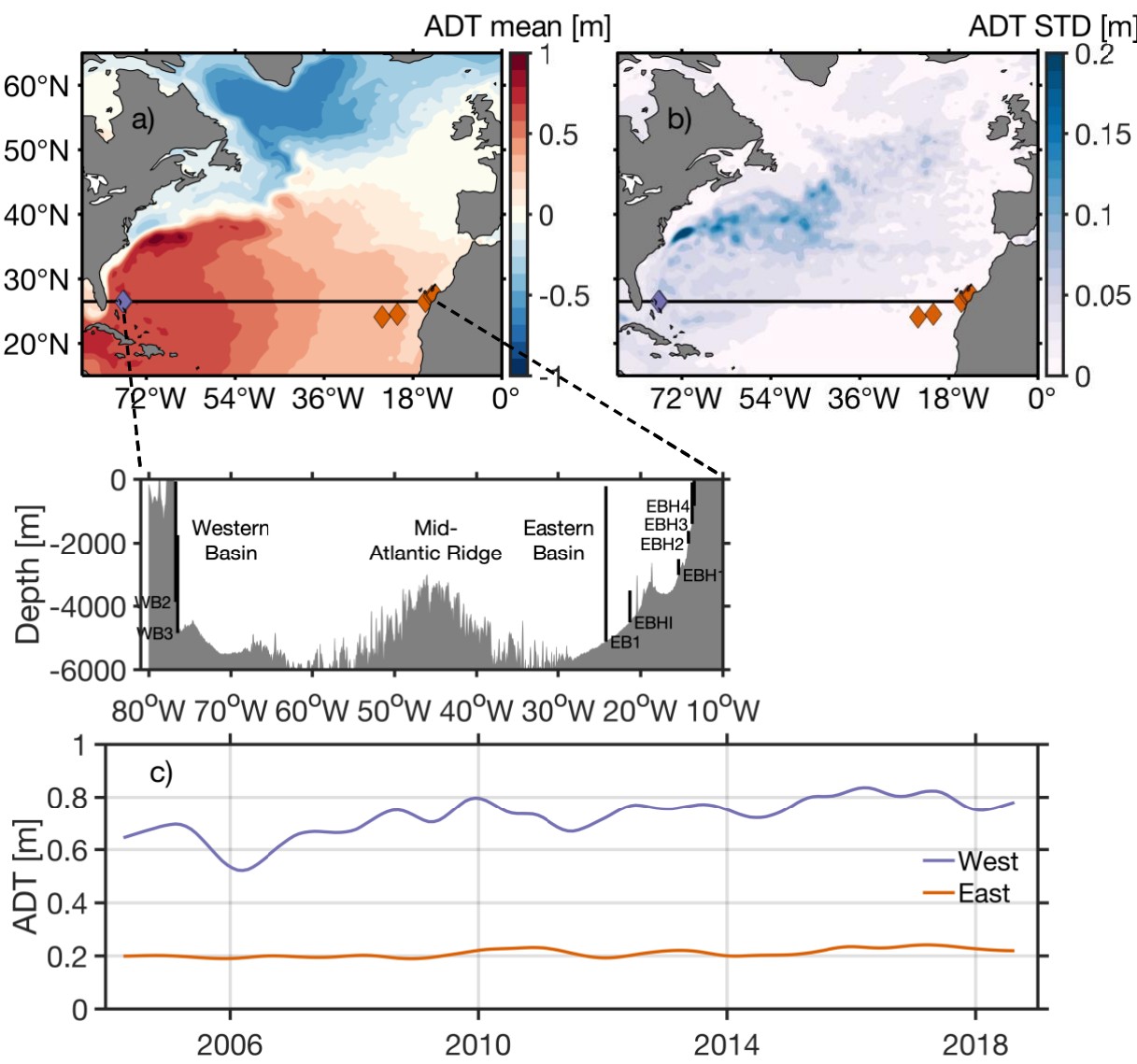

**Figure 1.** Absolute Dynamic Topography (ADT) (a) mean and (b) standard deviation (STD) across the North Atlantic over 2004 to 2018 time period. Western (purple diamonds) and eastern (orange diamonds) moorings are indicated along the RAPID array (black line). The inset in panel a) shows a vertical cross section of the basin with the location and label of RAPID moorings. (c) Time series of the western (purple line) and eastern (orange line) ADT near the locations of the western (WB2, WB3) and eastern moorings (EB), respectively. Units in m.

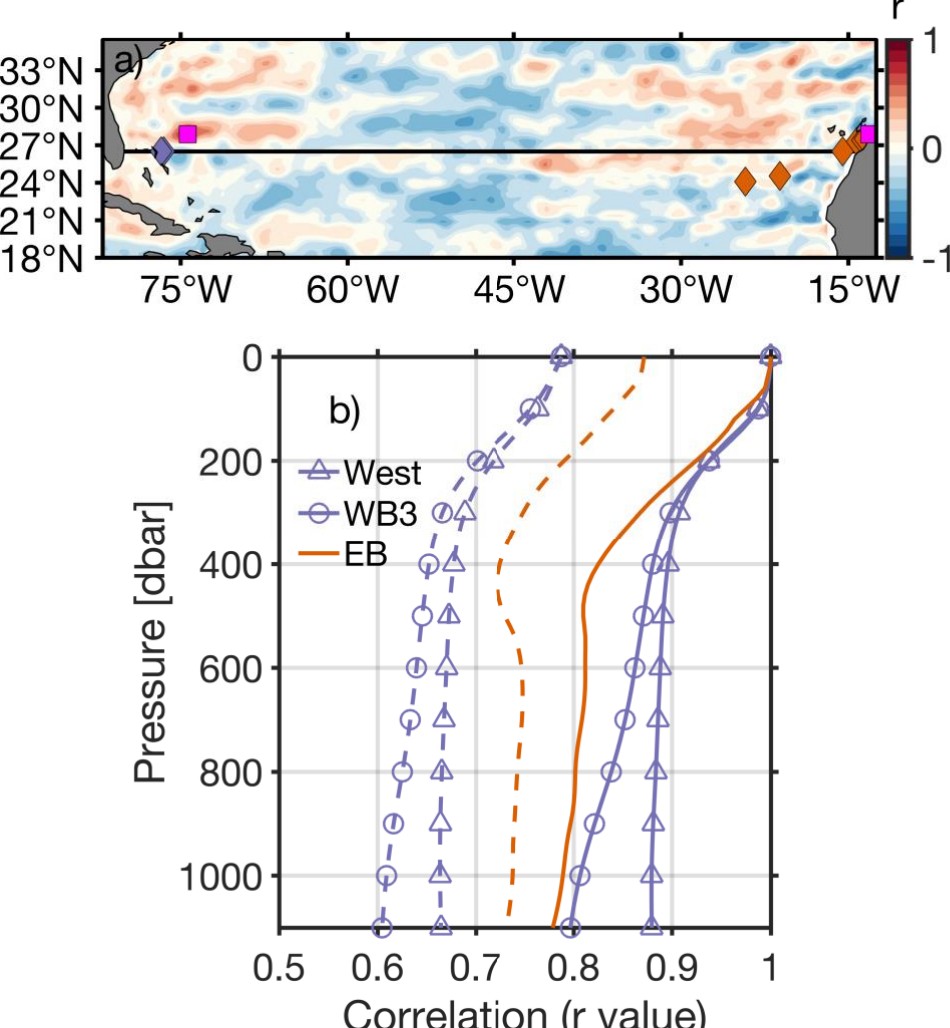

**Figure 2.** a) Correlation map between east-west difference in SLA ($\Delta\eta$) and the RAPID upper mid-ocean transport ($T_{UMO}$). Magenta squares in the western and eastern part of basin indicate region of maximum correlation between $\Delta\eta$ and $T_{UMO}$. Western (purple diamonds) and eastern (orange diamonds) moorings are indicated along the RAPID array (black line). b) Correlation between SLA and dynamic height from RAPID moorings West (purple triangle line), WB3 (purple circle line), and EB (orange solid line) at each pressure. Dynamic height from RAPID moorings is referenced to SLA at the surface ($\phi_\eta$). The dashed lines indicate the 95% significance level for SLA and dynamic height r-values per mooring using the two-tailed *t* test.


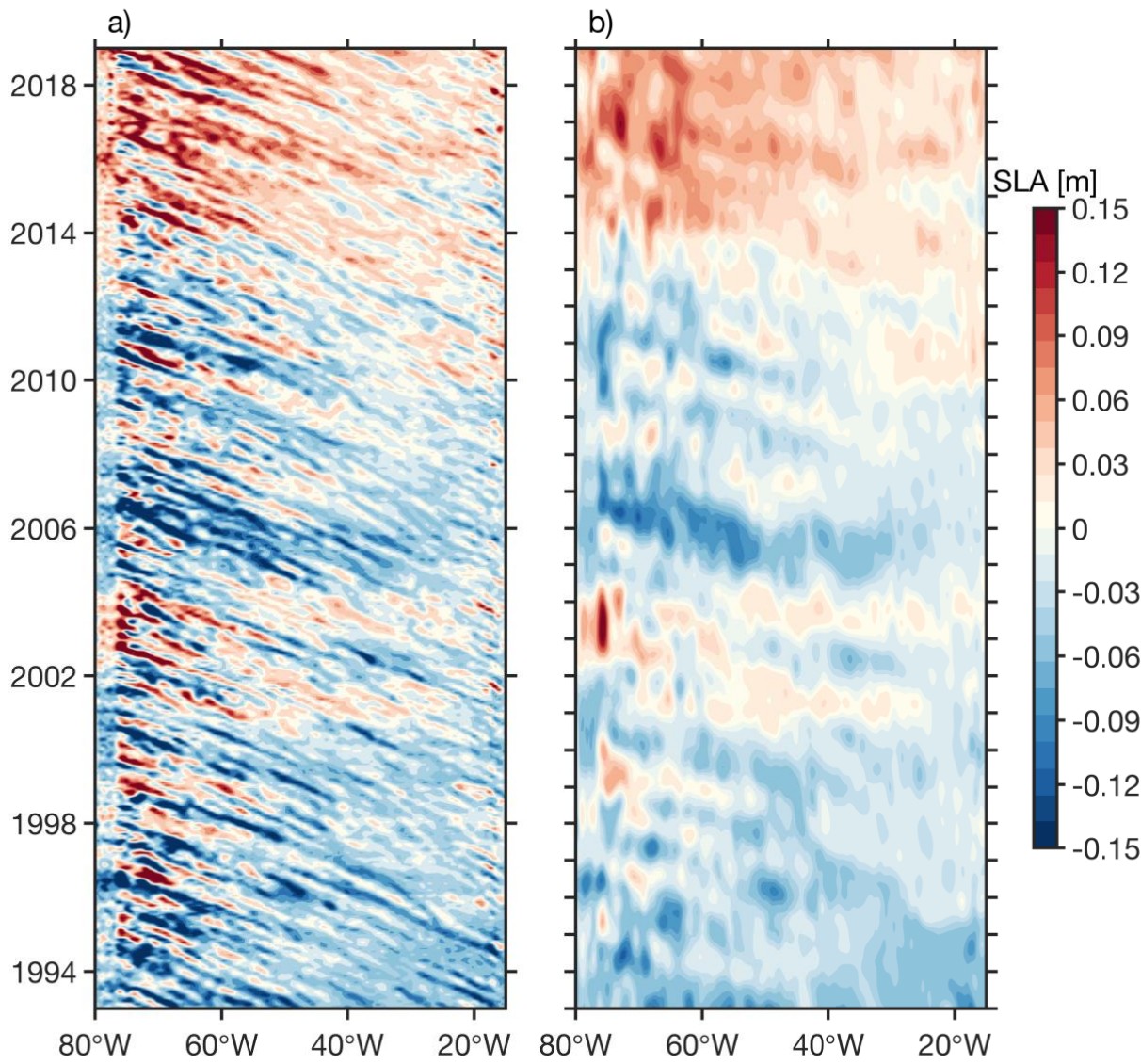

**Figure 3.** Hovmoller of sea level anomaly (SLA) at ¼° resolution in space at 26.625°N over 1993 to 2017 for monthly resolution (a) and 18-month smoothed (b). The annual seasonal climatology has been removed from the SLA. Units in m.

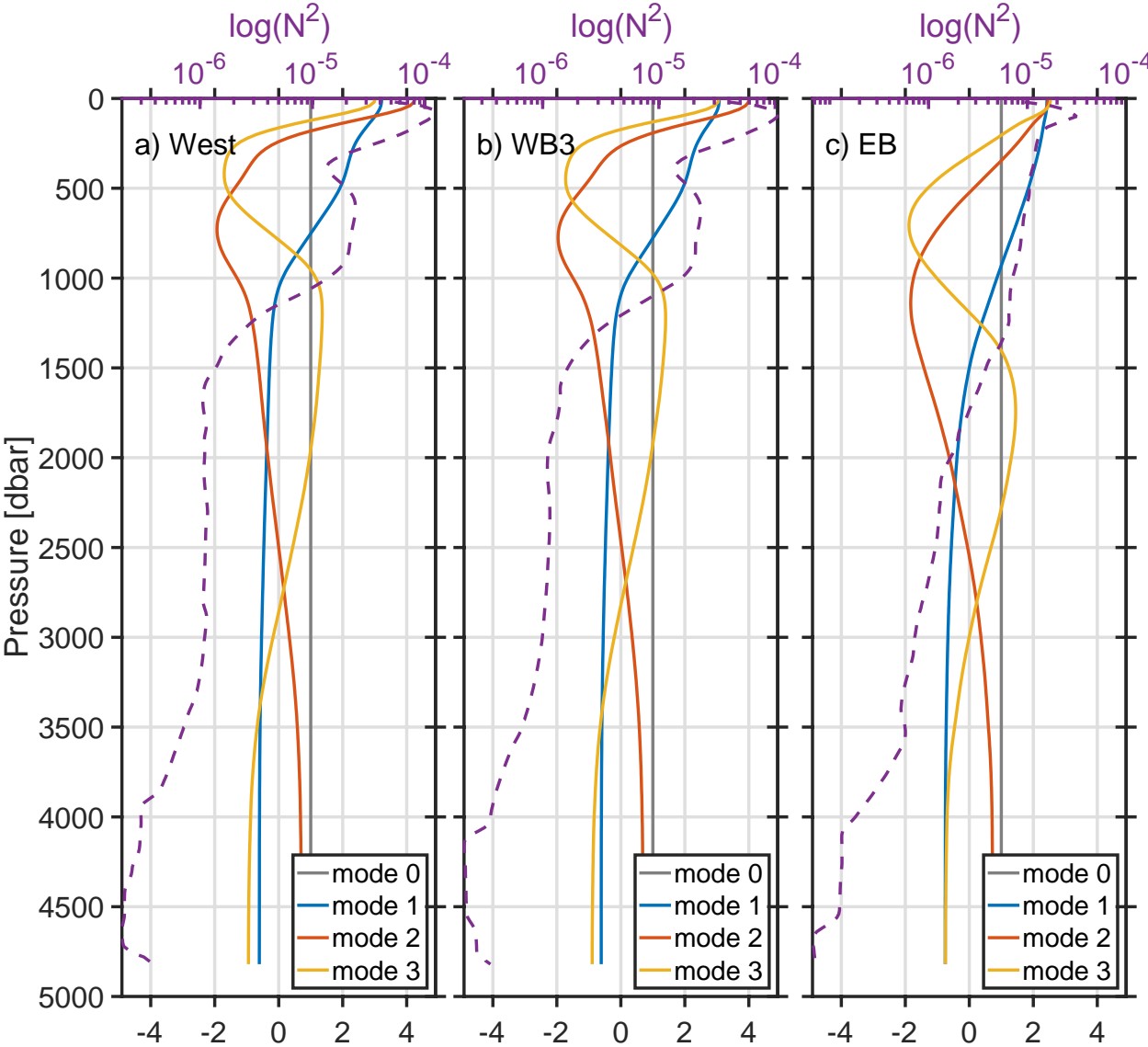

**Figure 4.** The buoyancy frequency ($N^2$; dashed purple line) and the horizontal velocity normal modes: the barotropic mode (i.e. mode 0; grey line) and the baroclinic normal mode 1 (blue line), mode 2 (orange line) and mode 3 (yellow line) at RAPID moorings: a) West, b) WB3, and c) EB. The modes have been scaled to satisfy Eq. (8) and are dimensionless.

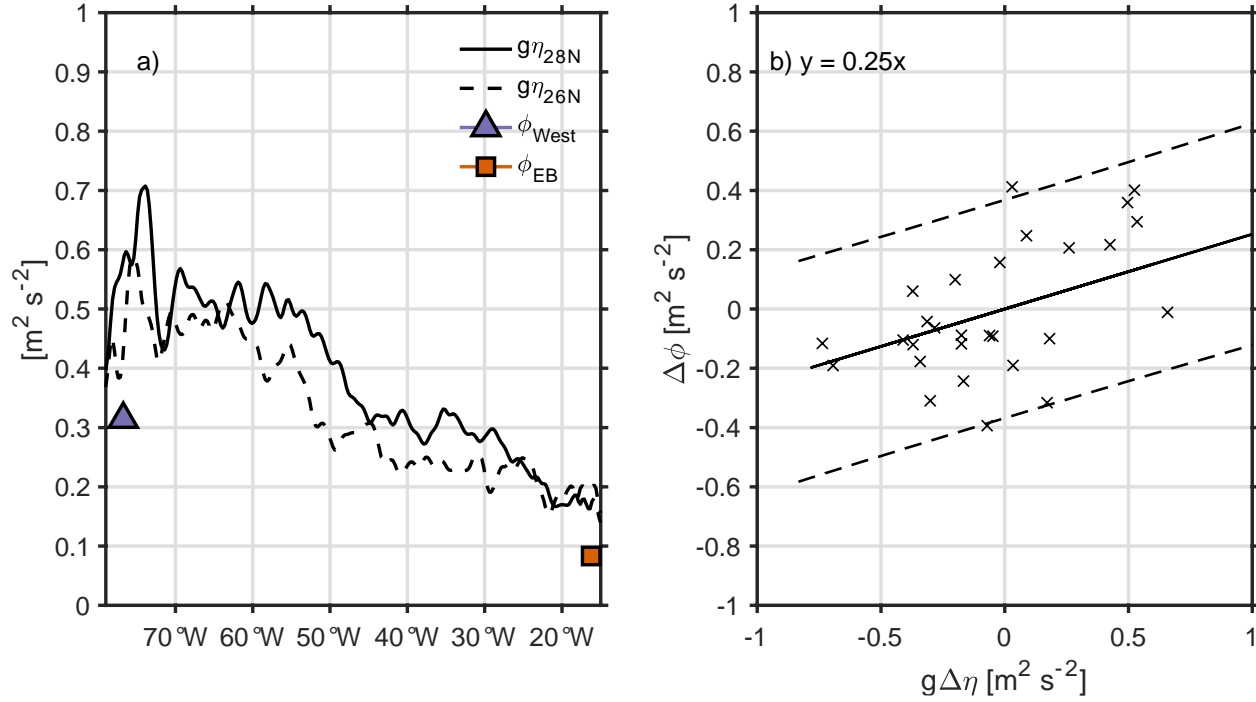


**Figure 5.** (a) The root mean square (RMS) at every longitude point of the SLA ($\eta$) at 26.125°N (black dashed line; latitude of RAPID moorings) and at 27.875°N (black solid line; latitude of maximum correlation between $\Delta\eta$ and $T_{UMO}$ – see Fig. 2). Symbols indicate the RMS of the dynamic height anomaly ($\phi$) at the surface (z = 0) at West

(purple triangle) and EB (orange square) moorings. (b) Scatterplot of the east-west difference in $\eta$, i.e. $\Delta\eta$, vs east (EB) – west (West) difference in the $\phi$, i.e. $\Delta\phi$, and best fit regression line (solid line). In (b) data are detrended and subsampled every 6 months. In both plots, $\eta$ is multiplied by gravitational acceleration. Dashed lines indicate the 95% confidence interval. Units in m$^2$ s$^{-2}$.


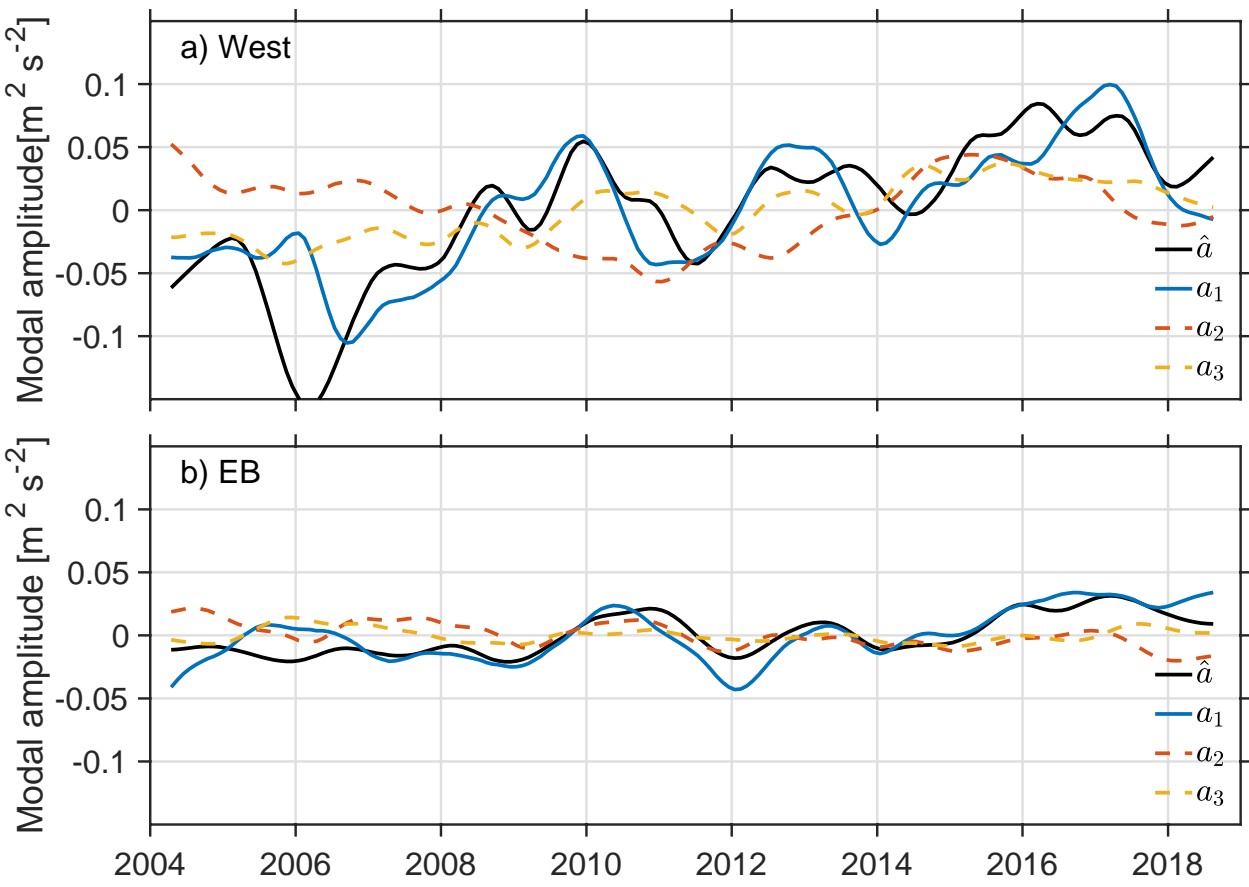

**Figure 6.** Modal amplitudes, $a_n$, fit to the dynamic height anomaly ($\phi$) and the baroclinic modes, $F_n$, as per Eq. (9) at (a) West and (b) EB moorings. The first three modal amplitudes: $a_1$ (blue line), $a_2$ (orange dashed line), and $a_3$ (yellow dashed line) are compared with modal amplitude estimated from SLA, $\hat{a}$ (black line) as per Eq. (10). Units in $m^2 s^{-2}$.

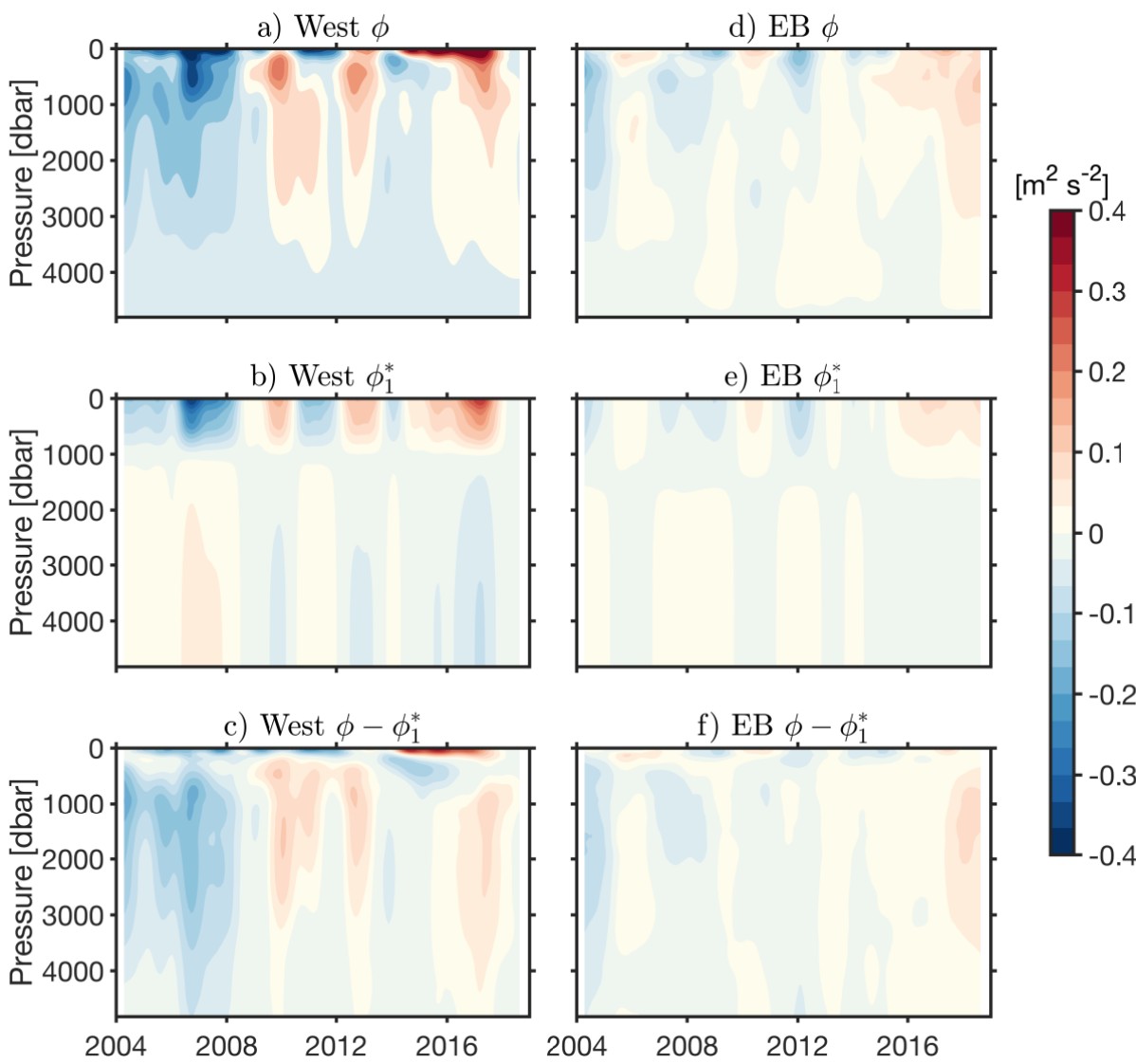

**Figure 7.** Dynamic height anomaly ($\phi$) from the surface to the reference pressure 4820 dbar over the 2004 to 2018 period from RAPID moorings at West (a) and EB (d). Reconstruction of the dynamic height anomaly ($\phi^*$) at West (b) and EB (e) using the first baroclinic mode ($F_1$) and the corresponding modal amplitude ($c_1$) as per Eq. (11). The difference between the $\phi^*$ for the first baroclinic mode and $\phi$ at West (c) and EB (f). Units in $m^2\,s^{-2}$.

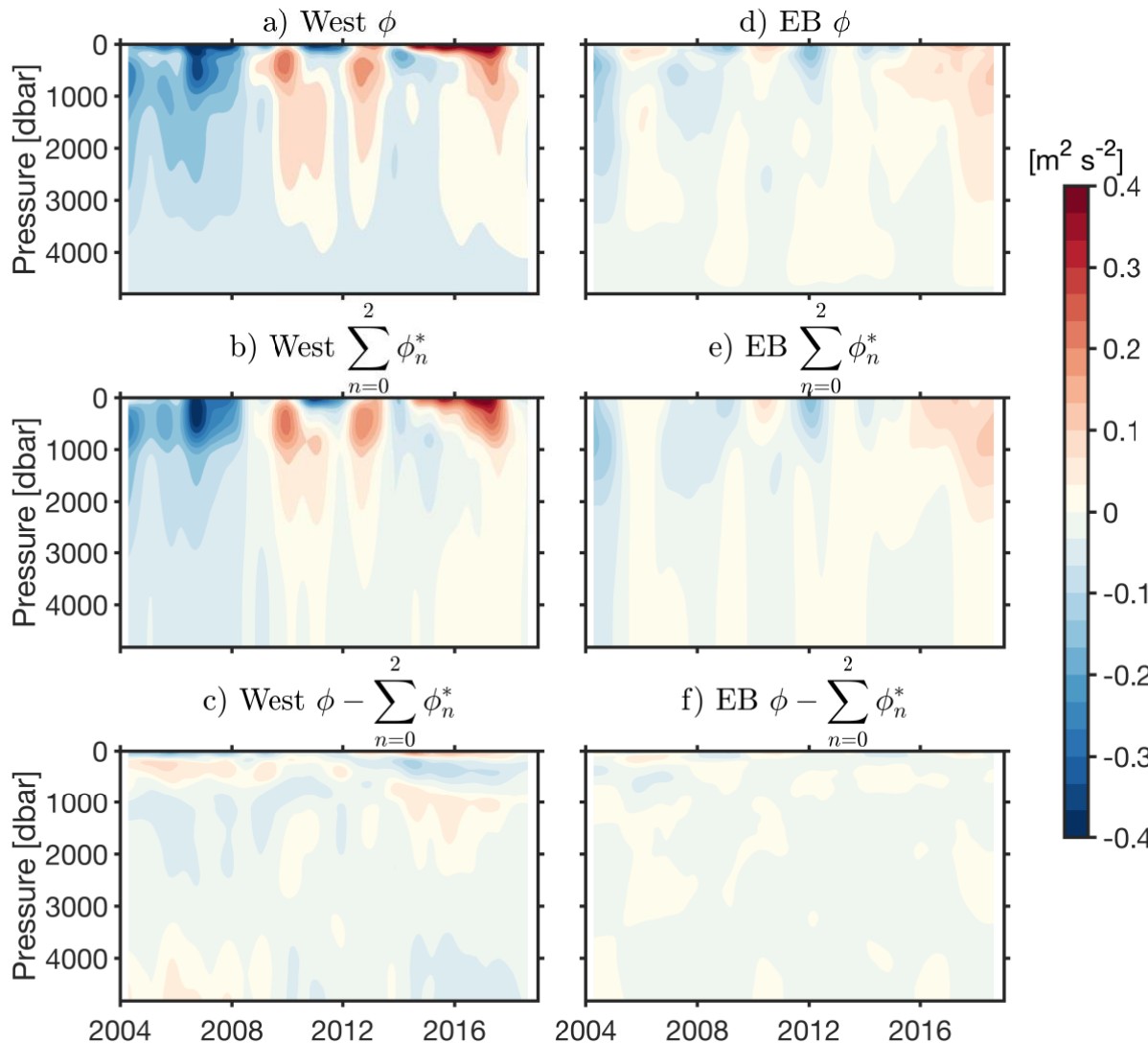

**Figure 8.** Same as in Fig. 7, except the contribution of the barotropic mode ($n = 0$) and the first two baroclinic modes ($n = 0,1,2$) are included in the dynamic height anomaly reconstruction ($\phi^*$) at West (b) and EB (e).


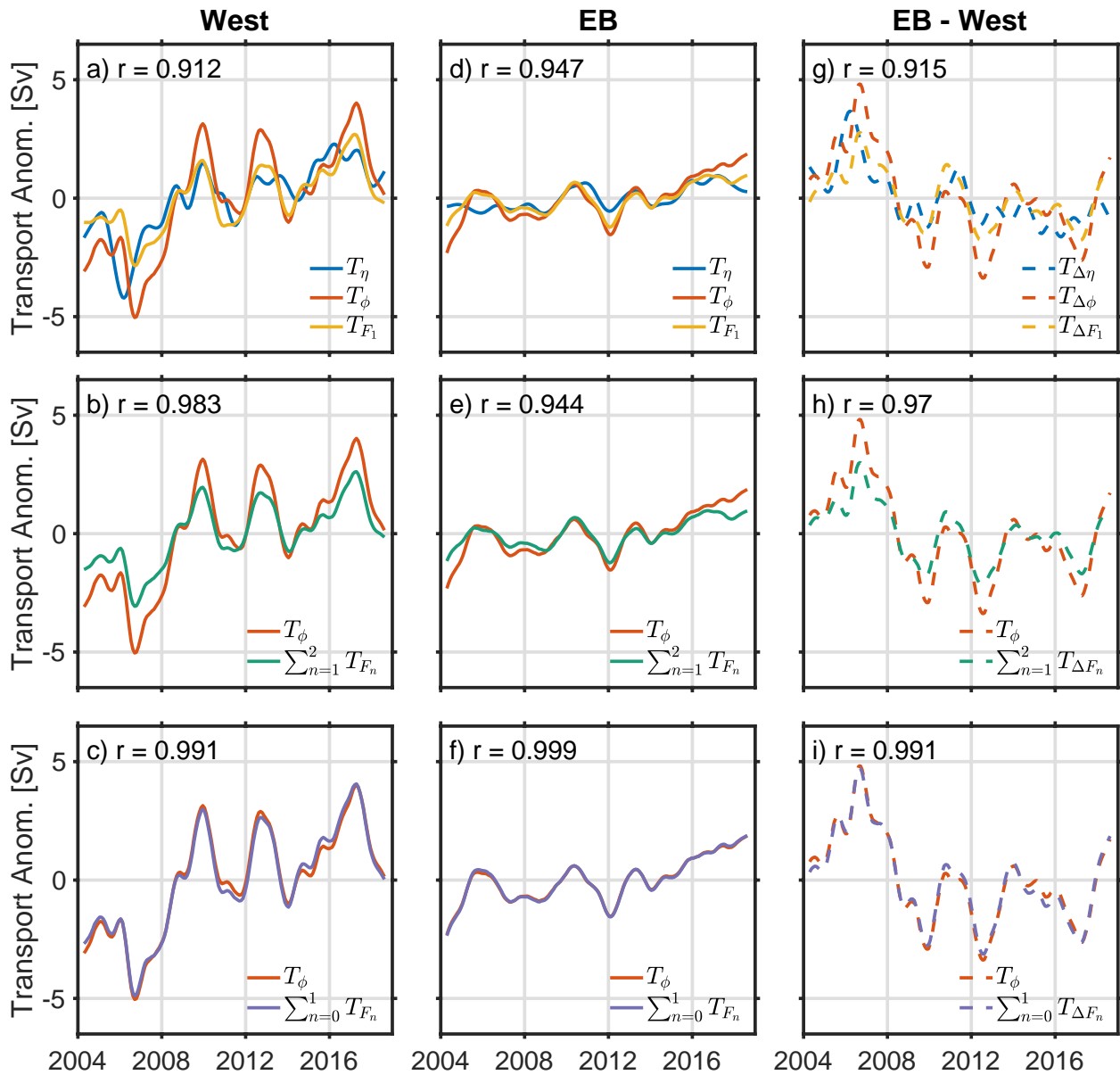

**Figure 9.** Transport anomalies estimated from dynamic height anomalies, $T_\phi$ (orange line), as per Eq. (12) at West (a,b,c), EB (d,e,f), and EB – West (g,h,i). Transport anomalies estimated from the reconstructed dynamic height anomaly, $T_{F_n}$ as per Eq. (13), for mode 1 (yellow line; a,d,g), modes 1 and 2 (green line; b,e,h), and mode 0 and 1 (purple line; c,f,i) at West, EB, and EB – West respectively. Satellite-based transport, $T_\eta$ (blue line), as per Eq. (14) is estimated in the (a) west ($\eta$ at 27.875°N and 74.375°W), (d) east ($\eta$ at 27.875°N and 13.125°W), and (g) east - west. Correlation between $T_\phi$ and $T_F$ indicated in each plot. Units in Sv.

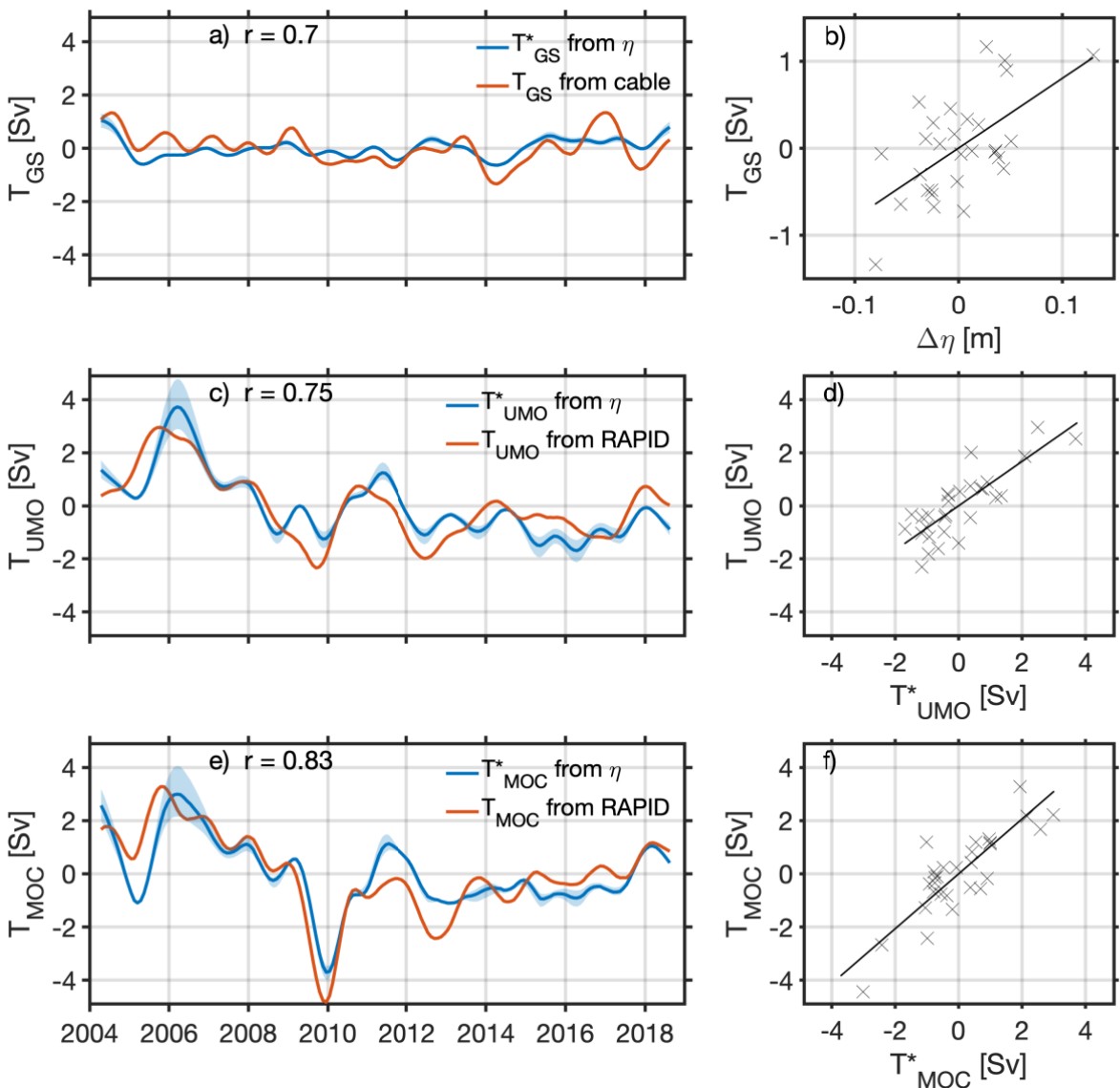


**Figure 10.** a) Time series of the Gulf Stream transport ($T_{GS}$) from submarine cables (orange line; Baringer and Larsen, 2001) and the satellite-derived $T^*_{GS}$ estimates as per Eq. (17) (blue line). b) Scatterplot between the east (27.625°N and 77.125°W) - west (27.625°N and 80.125°W) SLA ($\eta$) and $T_{GS}$ with the line of best fit linear regression – note y and x axis limits reduced to improve visibility of datapoints. c) RAPID upper mid-ocean transport ($T_{UMO}$; orange line) and the satellite-based $T^*_{UMO}$ transport estimates as per Eq. (18) (blue line). d) Scatterplot between the $T^*_{UMO}$ and $T_{UMO}$ with the line of best fit linear regression. c) RAPID upper mid-ocean transport ($T_{MOC}$; orange line) and the satellite-based $T^*_{MOC}$ transport estimates as per Eq. (19) (blue line). f) Scatterplot between the $T^*_{MOC}$ and $T_{MOC}$ with the line of best fit linear regression. Blue shading indicates uncertainty estimates using a Monte Carlo technique (a,c) as described in text, and blue shading in (e) is the sum of the confidence intervals from (a,c). In (b,d,f) data are subsampled every 6 months. Units in Sv.

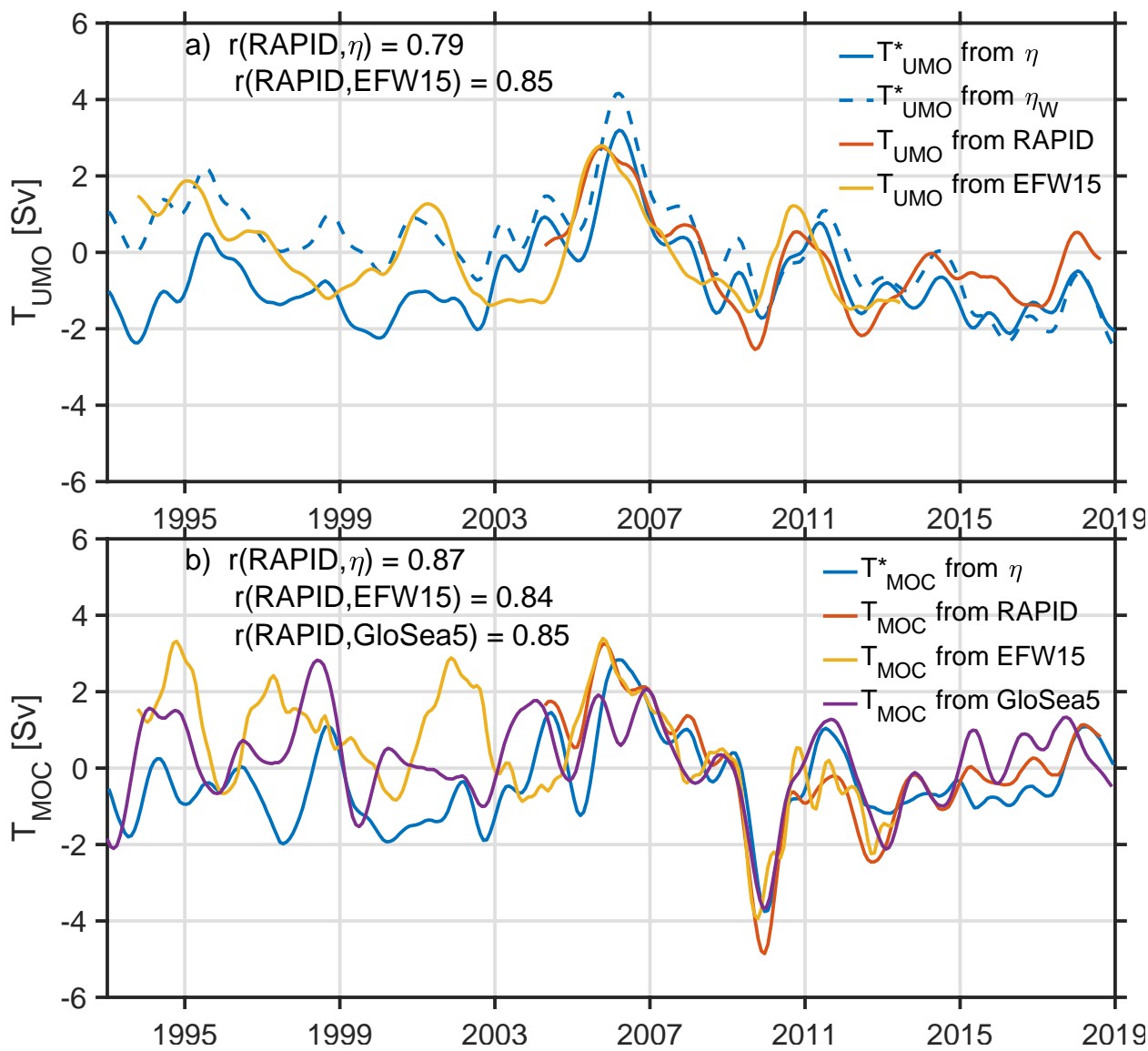

**Figure 11.** (a) The upper mid-ocean transport, $T^*_{UMO}$, from satellite ($\eta$) (blue line) as per Eq. (18), $T^*_{UMO}$, from $\eta$ whilst the eastern component is held constant (blue dashed line), $T_{UMO}$ from RAPID (orange line), and $T_{UMO}$ from EFW15 (yellow line). (b) MOC transport, $T^*_{MOC}$, from $\eta$ (blue line) as per Eq. (19), $T_{MOC}$ from RAPID (orange line), EFW15 (yellow line), and GloSea5 (purple line). Correlations indicated in graphs are over the period of overlap between the RAPID and EFW15 time series, i.e. 2004 to 2013. Units in Sv.