# Peer review of "A dynamically based method for estimating the Atlantic meridional overturning circulation at $26^{\circ}N$ from satellite altimetry"

_Ocean Science, 2021_

## Author Response (AR1)

Ref.: os-2021-10
A dynamically based method for estimating the Atlantic overturning circulation at 26N from satellite altimetry
* * *
**Reviewer #1**

Review of "A dynamically based method for estimating the Atlantic overturning circulation at 26°N from satellite altimetry" by Alejandra Sanchez-Franks, Eleanor Frajka-Williams, Ben I. Moat1, and David A. Smeed

This study examines a new dynamically-based technique for estimating the temporal variability of the AMOC and its constituents utilizing satellite altimetry, with the main goal of providing the base for efficient long-term backups for continuous estimates of the AMOC across different latitudes, as opposed to single mooring lines at key latitudes along the Atlantic Ocean. They focus their analyses on the RAPID/MOCHA/WBTS array at 26.5°N, by examining the vertical structure of the flow from the RAPID dynamic height moorings combined with satellite altimetry to evaluate a new method for estimating the AMOC variability at this particular latitude. The authors analyze the AMOC meridional transport variability from satellite altimetry and compare it with the transport estimates from the RAPID Array, they investigate the vertical structure of the circulation by analyzing the vertical baroclinic and barotropic modes from the moored measurements, and estimate the upper mid ocean transport and AMOC from historical altimetry data available since 1993 by developing a dynamically based method, which incorporates information on the full-depth vertical structure of the flows. The latter is the most novel aspect of the study.

There is a large amount of work involved, the methods are adequate, and the manuscript tackles a very important topic of research given the necessity of developing new strategies for cost-effective, long-term sustained monitoring of the AMOC. The analysis presented is of importance for the wider scientific community.

However, the manuscript is quite often imprecise, while the methods are adequate it is quite under referenced, and I think that a better flow between the different sections could be achieved. Most sections seem like separate studies and there is very little guidance for the motivation/background for each of the analyses. My general impression is that that the authors do not reach to a precise conclusion about the implementation of this new technique to evaluate AMOC variability at 26.5°N.

In general, the manuscript would benefit from more discussion of the results in the context of the previous studies to aid in the interpretation. It is also not clear if all of the analyses described in the current manuscript need to be presented to reach a final conclusion. Another major concern is that the manuscript needs revision to highlight the new aspects of this study and to provide a discussion on how (and why) their new satellite based dynamical method improves other previous AMOC estimates using satellite altimetry, or if that is not the case it should be clearly formulated, and to explicitly mention what is new in this study compared to the other studies. This will strengthen the outcomes of the paper. In particular, the summary and conclusions section read like a list of the results from the previous sections but providing very little interpretation on how meaningful/important these results are, and most importantly, an assertive statement is missing about the advantages/disadvantages of this new method compared to other previous studies. This section needs an interpretation of the implications of the results for long-term AMOC monitoring using satellite altimetry at 26.5°N (and possibly at other latitudes).

My overall recommendation is that the manuscript cannot be accepted in its present form but it may be accepted for publication after moderate to major revisions.

We thank the reviewer for their constructive comments.

We have now revised the manuscript to include further background information and discussion from previous studies to give better context to results from this work. We have also re-written some sections to add clarity on the novelty of this work and its advantages and disadvantages compared to previously published methods.

A supplementary materials section has now been added with new analysis showing the results of sensitivity testing depending on the choice of Gaussian filtering of the data on the results (Fig. S1, S6, S7, S8). An additional figure comparing SLA to dynamic heigh anomaly thickness has been added to provide a more complete discussion of the relationship between SLA and dynamic height (Fig. S2) in section 3. Further discussion of the scale factor a new figure showing east-west difference in dynamic height vs SLA (Fig. S3) has been added to section 4.3. A new figure showing transport anomalies from the reconstructed dynamic height anomaly for the first baroclinic mode, the second baroclinic mode, and the barotropic mode has been added and discussed in section 4.4 (Fig. S4). Finally, a new figure showing the correlation map between the SLA in the Florida Straits and the cable-based Gulf Stream transport (Fig. S5) is added and discussed in section 5.

Details for all these changes are described in the reply to specific comments below. All reviewer comments are itemized and addressed in blue. Line numbers given here are in reference to the tracked changes manuscript. New text added to the manuscript is in red.

Please find specific comments below:

1. Title: I suggest adding "meridional" between "Atlantic" and "overturning" to be more specific.

Thank you for this suggestion. The title has been amended to include meridional.

2. Lines 1-2: Here and throughout the text. I suggest not using "surface" when referring to the upper 1000 m. While this may be somehow a commonplace for the AMOC community this could be potentially confusing for other readers/communities. Especially in the context of this paper this is confusing because you also refer to "surface" velocity for the altimetry estimates at the ocean's surface (e.g., Line 51). Maybe just use "upper 1000 m" when you refer to the 1000 m AMOC layer carrying northward flow.

As suggested, "surface" has been replaced with "upper" throughout the text, wherever appropriate.

3. Line 7: I suggest changing "High global" for "Near-global".

This change has been made.

4. Line 10: Here and throughout the text, do you mean that the method is valid only for 18-month periods or also for time scales longer than 18 months? This should be clear here and elsewhere.

This method is valid for 12- to 18- month smoothing, in that it faithfully reproduces the observed transport variability at 26N after lowpass filtering the high frequency (sub-annual) variations. We have added sensitivity testing to demonstrate the equivalent transport fluctuations estimated using shorter windows (see supplementary figures S1, S6, S7, S8). This is also discussed in sections 2, 3, and 5, and see replies to comments 27, 37, and 52. However, for longer timescale variability (e.g. decadal timescales), we do not have sufficiently long records to validate the method.

5. Lines 10-11: I think that the "compares well" statement is arguable. I suggest this information to be provided as how much variance is recovered by the satellite estimates for the different AMOC constituents rather than providing the correlation coefficient. For instance for UMO the variance recovered from the altimetry product is about 56% and for the Gulf Stream transport is 50%. This leaves a considerable portion of the variance unexplained by the satellite-based estimates. But for the full AMOC the variance increases to 64% (the reason for the increased explained variance by ~8-14% for the full AMOC estimate is something that should be discussed in the Results section). I feel this is much more useful information for the reader.

Thank you for this suggestion. This change has been made: the abstract now cites variance captured rather than correlation coefficients in L19: Here we show a direct calculation of ocean circulation from satellite altimetry of the upper mid-ocean transport (UMO), the Gulf Stream transport through the Florida Straits (GS), and the AMOC using a dynamically based method that combines geostrophy with a time mean of the vertical structure of the flow from the 26°N RAPID moorings. The satellite-based transport captures 56%, 49%, and 69% of the UMO, GS, and AMOC transport variability, respectively, from the 26°N RAPID array on interannual (18-month) time scales.

6. 12-13: Is it really 17% variance for the barotropic mode? What about the other BC modes?

We have clarified in the abstract in L24 of the abstract and in the results section that: Further investigation into the vertical structure of the horizontal velocity shows that the first baroclinic mode accounts for 83% of the interior geostrophic variability, the barotropic mode accounts for 82% and the combined barotropic and first baroclinic mode account for 98% of the variability.

We have also added a new figure to the supplementary materials (Fig. S4) showing the individual contributions from the barotropic and the second baroclinic components. This is now discussed in section 4.4.

7. Lines 14-15: In the abstract these lines are the only reference to the method that gives the title to the manuscript "Finally, the UMO and the AMOC are estimated from historical altimetry data (1993 to 2018) using a dynamically based method that incorporates the vertical structure of the flow." But there is no information about the outcome of this analysis. I suggest revising in light of the novel aspects of this study compared to other studies.

We thank the reviewer for this suggestion. We have revised this paragraph to show main results from the dynamic method earlier on in the abstract (also see reply to comment 5) and accordingly re-written the sentence on the historical reconstruction in L16: Near-global coverage of surface ocean data from satellite altimetry is available since the launch of TOPEX/Poseidon satellite in 1992 and has been shown to provide reliable estimates of surface ocean transports on interannual time scales including previous studies that have investigated empirical correlations between sea

surface height variability and the overturning circulation. Here we show a direct calculation of ocean circulation from satellite altimetry of the upper mid-ocean transport (UMO), the Gulf Stream transport through the Florida Straits (GS), and the AMOC using a dynamically based method that combines geostrophy with a time mean of the vertical structure of the flow from the 26°N RAPID moorings. The satellite-based transport captures 56%, 49%, and 69% of the UMO, GS, and AMOC transport variability, respectively, from the 26°N RAPID array on interannual (18-month) time scales. Further investigation into the vertical structure of the horizontal velocity shows that the first baroclinic mode accounts for 83% of the interior geostrophic variability, the barotropic mode accounts for 82% and the combined barotropic and first baroclinic mode account for 98% of the variability. Finally, the methods developed here are used to reconstruct the UMO and the AMOC for the time period pre-dating RAPID, 1993 to 2003. The effective implementation of satellite-based method for monitoring the AMOC at 26°N lays down the starting point for monitoring large-scale circulation at all latitudes.

8. In general, there should be a hyphen between "long" and "term".

Done.

9. Line 29: I don't think it is necessary to add "m3 s-1" after volume, or perhaps add Sv. Please add "and heat" before transport and add a reference to Johns et al. (2011) (and/or newer reference if available).

Units for volume are now removed, mention of heat transport and Johns et al (2011) reference now added.

10. Line 33: moorings are deployed on both flanks of the Mid-Atlantic Ridge, this is not clear in the text.

This is now clarified in text in L68: placing moorings in the western and eastern endpoints of the Atlantic basin, as well as both flanks of the mid-Atlantic ridge,…

11. Line 34-35: This part of the sentence "interior flow going through them" is not clear. I suggest "basin-wide interior flow". Please add key references (e.g., Cunningham et al 2007; Kanzow et al., 2010; McCarthy et al. 2011).

Sentence amended to include clarification as suggested. References suggested have been added as well (L70-71).

12. Line 35: I don't think that a basin-wide density field is extracted from the rapid moorings but rather the vertical density structure at each mooring site from which the density field can be somehow estimated at the western and eastern parts of the array and to some extent near the Mid-Atlantic-Ridge.

Thanks to the reviewer for pointing this out. The text has been amended accordingly in L71.

13. Line 36-37: Please add a few words about the times scales for these Rossby-wave like features.

The text has now been amended to include in L74: seasonal and subannual time scales (Hirschi et al., 2007).

14. Lines 38-40: While this work is focused only in the North Atlantic MOC at 26.5°N, I think it is important to provide key references for the other international efforts to measure the AMOC from in situ arrays at other single latitudes as this is important for the main objective of this study. Frajka-Williams et al. (2019) provide a very nice review of all the arrays but at least one key reference for each of the basin-wide observing arrays should be included. I suggest: Lozier et al., (2019) for OSNAP in the subpolar North Atlantic, Hummels et al. (2015); Herrford et al. (2021) for the TRACOS/Tropical MOC array at 11°S, and Meinen et al., (2018); Kersalé et al. (2020) for the SAMOC/SAMBA array in the South Atlantic at 34.5°S. It may also be worth including a brief statement here about the limitations for studying meridional coherence of the AMOC for instance based on the large meridional distances among the arrays to provide more context for the goals of your study.

Key references for other international arrays monitoring the AMOC, as well as limitations of the array, now given in L75-85: Other international efforts to measure the AMOC via mooring array programmes include: SAMBA (South Atlantic MOC/South Atlantic MOC Basin-wide Array) at 34.5°S (Meinen et al., 2018; Kersale et al., 2020), TRACOS (Tropical Atlantic Circulation and Overturning) at 11°S (Hummels et al., 2015; Herrford et al., 2021), and OSNAP (Overturning in the Subpolar North Atlantic Program) in the subpolar North Atlantic (Lozier et al., 2019). RAPID and these other mooring array programmes have made step-change advancements in our understanding of the AMOC over the last 15 years (Frajka-Williams et al., 2019), however they are limited to a single line of latitude and alone cannot be used to infer upstream/downstream changes in the larger scale structure of the AMOC. Further, the large meridional distances between the arrays (e.g. 34.5°S, 11°S, 26°N, 58°N), combined with the relatively short length/lifespan of the programs (e.g. the oldest is RAPID, initiated in 2004), preclude our ability to understand AMOC connectivity, i.e. its meridional coherence, which is still poorly understood (e.g. Bingham et al., 2007).

15. Line 41: I suggest changing "Altimetric measurements collect data…" -> satellite altimetry measures data…

Done.

16. Line 41-42: Change "means" for "mean" or "tool". This sentence requires references and a more extensive explanation or discussion about the limitations of using SSH or satellite altimetry for estimating boundary flows given their relevance for accurate AMOC estimates.

Done. Further discussion on the limitations of satellite altimetry is also give, see replies to comments 17, 19, 20.

17. Lines 46-50: Please provide more information about time scales, percentage of variance explained by the proxies developed in previous studies, as these are really important aspects when computing velocities/transports from altimetry or other ancillary data with lower temporal resolution than that provided by the in situ continuous-in-time mooring measurements.

The text has been amended to include details on time scales, percentage variance, etc.. in L100-109: For example, EOF analysis of satellite sea level anomaly (SLA) suggested altimetry could be useful indicator for AMOC transports along 40-50°N on interannual time scales (Bingham & Hughes, 2009). Frajka-Williams, (2015) found that the relationship between satellite SLA and

RAPID dynamic height was robust enough to be used to create a proxy for the upper mid-ocean transport, which when combined with Ekman transport (derived from reanalysis products) and Gulf Stream transport (from a submarine telephone cable) accounted for more than 90% of MOC variability on interannual time scales. The method developed in Frajka-Williams (2015) was purely statistical. The relationship between dynamic height from RAPID moorings and satellite SSH at 26°N was also found to be robust (accounting for up to 72% of variance at a station 500 km offshore) (Kanzow et al., 2009), however this relationship was found to deteriorate in proximity to the western boundary (Bryden et al., 2009; Clément et al., 2014).

18. Line 48: I think it is important to mention on which time scales.

The sentence is now revised to specify the time scales are interannual in L106.

19. Line 52: "While this is enough to infer AMOC variability (due to the baroclinic nature of the surface intensified ocean)". This sentence is not clear, please revise. AMOC variability on which time scales? Please provide a key reference for this statement.

The sentence is now revised as suggested in L111-116: One of the limitations of satellite altimetry is that it only provides information about the sea surface. The information captured by satellite altimetry at the surface is representative of variability associated with the first baroclinic mode (Hirschi et al., 2007). Because of the baroclinic nature of the surface intensified ocean, Rossby wave theory suggests that some MOC variability can be linked to satellite altimetry on subannual (Hirschi et al., 2007) and subannual to interannual (Hirschi et al., 2009) time scales. However, satellite altimetry cannot tell us anything about the vertical shape of the horizontal velocity which forms part of the AMOC structure.

20. Line 54: Please, change "some success" to more precise information about the results from previous studies and on what is new in this study and how this studies has improved the previous efforts.

More precise information is now given on results from previous studies in L118-131: In particular, Hirschi et al. (2009) found that combining SSH with baroclinic structure, using a numerical model, yielded a robust estimate of eastern and western branches of the MOC, but results could not be replicated for reconstructing variability of the basin-wide MOC on subannual to interannual time scales (Hirschi et al., 2007; Hirschi et al., 2009). They found that even small errors in the western and eastern components could overwhelm MOC variability, and only highly accurate estimates of the western and eastern branches would yield a reliable reconstruction of the basin-wide MOC. In a study focused on the impact of eddy dynamics on basin-wide transport, Kanzow et al. (2009) tested whether geostrophic transport could be obtained by combining geostrophic surface flow from SSH with the dominant vertical mode of horizontal velocity; they found that the correlation between the difference in SSH at eastern and western endpoints of the basin had little skill in capturing the upper mid-ocean transport and suggested the reason for this was not enough satellite data close to the shelf as well as changes in the vertical structure of the flow moving toward the western boundary. Szuts et al. (2012) similarly found that SSH captured fluctuations in the centre of the basin but struggled to capture the variability closer to the boundaries, where the vertical structure of the flow was not as well described by the dominant baroclinic mode.

21. Line 55: Please add "in the North Atlantic at 26.5°N" at the end of the sentence since that is the latitude you are exploring and the results may differ for different latitudes.

Done.

22. Lines 55-60: I strongly suggest highlighting the new aspects of this study here.

New aspects of study now highlighted in L133-159: The length of the RAPID records has more than doubled since the aforementioned studies that combined geostrophy and vertical structure of horizontal velocity (e.g. Hirschi et al., 2009; Kanzow et al., 2009; Szuts et al., 2012), so the time is right to re-examine the issues (e.g. problems at the boundary, including contribution from vertical modes) and build on the methods developed in these previous studies. Thus, here we combine geostrophic principles with the vertical structure of the horizontal flow to develop a satellite-based MOC transport on interannual time scales. In contrast to earlier efforts by Hirschi et al. (2009) and Kanzow et al. (2009), the longer records now available at RAPID enable us to test the methods for longer timescales (interannual, rather than sub-annual), and we find that the skill is generally higher at interannual timescales than at shorter timescales. Unlike Frajka-Williams (2015), this method now uses geostrophy (east minus west differences in sea level anomaly) rather than finding the point location where the sea level anomaly has the strongest correlation with the transport variability. This change in the method is a prerequisite to developing similar methods at other latitudes as it reduces (though does not entirely eliminate) the requirement for in situ data to 'train' the method. In addition, while Szuts et al. (2012) and Clement et al. (2014) found that multiple modes are required to explain the variance in the western boundary profiles, we show that on interannual timescales, the contributions from higher modes are reduced and the first mode explains a majority of the variance.

23. Section 2-5: In general throughout all the results sections, I suggest highlighting what is different/new in this work compared to the previous studies (e.g., Kanzow et al 2009; Szuts et al., 2012; Frakja-Williams et al., 2015; Volkov et al., 2020).

New text highlighting key results from this study now incorporated in the introduction (see replies to comments 20, 22), throughout the rest of the text and the conclusions (see replies to comments 53).

24. Line 73: Without providing information about the vertical distribution of the observations in the array this is hard to follow and not very informative. As this information is relevant for the modal decomposition at the EB (Szuts et al., 2012), please clarify here that there are a set of moorings, which are concatenated to yield a single profile (Chidichimo et al., 2010).

Information about the basin-wide distribution of the moorings is now provided in Figure 1. Clarification on the how moorings have been concatenated into a single profile as now been added as well in L194 and L199: The western boundary used in the RAPID AMOC calculation includes moorings WB2, WBH2 and WB3 (concatenated into a single profile and hereafter referred to as West). When comparing with the SLA, data from the RAPID moorings WB2 (26.5°N and 76.75°W) and WB3 (26.5°N and 76.5°W) and West are used for the western component, and an amalgamation of data from the eastern boundary moorings between 13.75°W-24.22°W and 23.7-27.9°N (concatenated into a single profile (Chidichimo et al., 2010; McCarthy et al., 2015) and hereafter referred to as EB) are used for the eastern component (section 3 and 4) (Fig. 1a).

25. Lines 75-76: Please revise this sentence. It does not clearly formulate how dynamic height is computed from the T/S measurements and how the MicroCATS attached to the moorings provide continuous-in-time CTD measurements at discrete depth levels, or please provide a reference where the methods are explained. This is better explained in Line 160, probably part of that text could me moved here.

As suggested, information originally in L160 is now moved to L178 in methods section 2.1, and further detail is added to clarify how dynamic height is computed (L178): Temperature and salinity from individual MicroCAT records are vertically interpolated using the climatological profiles of dT/dp and dS/dp to produce a regularly-gridded profile of temperature and salinity on a 20 dbar grid (Johns et al. 2005; McCarthy et al. 2005). Dynamic height ($\phi$) is then calculated from the gridded density profiles as:

$$\phi(\text{p}) = \int_{4820}^{p} \delta\left(p'\right) dp',$$
(1)

where the specific volume anomaly ($\delta$) is defined as $\delta = \frac{1}{\rho}$.

Further details on method and instrumentation are found in McCarthy et al. (2015) as indicated in L200.

26. Line 83. Please add "after Kanzow et al., (2007)" at the end of the sentence.

Done.

27. Line 90: It is not clear why the authors choose 18 months. How sensitive are the results to the choice of the 18-month period for the Gaussian filter? Please add a brief discussion.

Further information and rationale for using an 18-month filter is given in L210-216: The filter is a Gaussian weighted moving average filter, where endpoints are computed such that when the number of points available is fewer than the window size, the window is resized/truncated and the average is taken from the elements available in the truncated window. Clément et al. (2014) found that Rossby waves and eddies are important contributors to geostrophic transport on time scales of 3-8 months and Kanzow et al. (2009) found that eddies do not affect AMOC on low frequency (interannual to decadal) time scales, hence a 18-month filter is deemed appropriate for removing the influence of eddies.

Additionally, we have now added some sensitivity tests to the results (section 3) and in the supplementary materials using 6 month, 12 month, and 24 month filters in addition to the 18 month filter. For details on the sensitivity results, please reply to comment 37 and 52.

28. Line 92: Please provide threshold values of the correlation coefficient r for the significance of the correlations at the 90 and 95% confidence limit.

When computing correlations, the linear trend from 2004-2018 is removed from the respective time series, and the statistical significance is determined using the effective degrees of freedom (DOF) calculated as the length of each time series over the integral time scale of decorrelation (Emery and Thomson, 2001). As the DOF and the confidence interval is taken into account to determine the significance we feel there is no need to state the confidence intervals here.

Please also see reply to comment 47.

29. Line 96: Please add more references for the methods to estimate transports from the RAPID array, especially please refer to the initial papers were the methodology was developed (e.g., Kanzow et al., 2010).

As suggested further references on initial methodology added in L225: Kanzow et al. (2007), Kanzow et al., (2010), Johns et al., (2011), Rayner et al., (2011), McCarthy et al., (2015).

30. Line 99: Please add a reference to previous calculations using the Florida Straits cable measurements (e.g., Meinen et al., 2010)

As suggested, Meinen et al., (2010) is also added to the Baringer and Larsen (2001) reference.

31. Equations (1)-(3): please provide the limits for the integrals (I assume they are not identical for each calculation).

Limits for integrals now included in all equations.

32. Line 114: Please define the gravitational acceleration g (not defined before) for Eq. (4)

Gravitational acceleration is defined in L264 for equation (4).

33. Line 126-133: Please provided more references for the key circulation aspects in the region. The results need to be discussed and put into context of previous observations.

Further references and text added in L289-290, and L295-297.

34. Line 132: add "the" before "RAPID array"

Done.

35. Line 134: and other places: sometimes the latitude of the RAPID array is referred as 26°N and sometimes to 26.5°N. Please pick one and be consistent throughout the text.

Done. 26N replaced 26.5N where appropriate.

36. Line 135: Please define "rms" as "root-mean-square error" when it is first mentioned (I am aware you define RMS below, but it should be defined here as well).

Done.

37. Lines 142-143: Are these time series filtered? This should be clear in the text.

Yes, as mentioned in methods section (2.1), all data has an 18-month Gaussian filter. Further clarifications have been added in L209-210 (section 2.1): In this study, all 12-hour and daily data are monthly averaged, the annual seasonal climatology is removed and then smoothed with a 18-month filter over the available time periods, except where indicated otherwise.

As well as in section 3.1 L313-314: RAPID $T_{UMO}$ and $\eta$ have had the annual seasonal climatology removed and an 18-month Gaussian filter applied.

Further, a new figure in the supplementary materials (Fig. S1) as well as discussion on sensitivity of results to the choice of filtering window is now added in L332-340: Further, sensitivity of the correlation between RAPID $T_{UMO}$ and $\Delta\eta$ to the choice of filter was tested for 6 month intervals at 6 month, 12 months and 24 months, in addition to the 18 month Gaussian filter (Fig. S1). The pattern of correlations between RAPID $T_{UMO}$ and $\Delta\eta$ across the subtropical North Atlantic

remained generally consistent between the four different choices of filter (as described above for the 18 month filter), where correlations decreased (for eastern SLA gridpoint located at 27.875°N and 13.125°W and the western SLA gridpoint located at 27.875°N and 74.375°W) and were lowest at r = 0.48 (statistically significant at 95% level) using 6 month filtering, increasing to r = 0.67 using 12 month filtering, and r = 0.79 when using the 24-month filter, contrasted with r = 0.74 when using the 18-month filter.

[Figure]

**Figure S1.** Correlation map between east-west difference in SLA ($\Delta\eta$) and the RAPID upper mid-ocean transport ($T_{UMO}$), where data has a 6 month Gaussian filter (a), a 12 month Gaussian filter (b), an 18 month Gaussian filter (c), and a 24 month Gaussian filter. Western (purple diamonds) and eastern (orange diamonds) moorings are indicated along the RAPID array (black line).

Also see reply to comments 27 and 52.

38. Lines 156-158: Please see my comment above. The definition of upper ocean (0-1000 m), surface, sub-surface should be carefully revised and consistent throughout the paper. Please also revise Lines 172-173.

Done for L156-158 and throughout rest of manuscript. L172 and L173 revised as well.

39. Line 165-166: The updated correlation between the dynamic height from he moorings and SLA should be discussed in the context of the previous studies and also some interpretation is needed. How does a longer mooring record improve the correlation?

Further analysis of the relationship between the SLA and the dynamic height, context from previous studies and interpretation have now been added to this section. The new analysis is a plot in the supplementary materials (Fig. S2) showing dynamic height anomaly thickness (0 -1100 dbar) which provides a measure of shear independent of the reference level. Corresponding text is added in L361-384: This is in agreement with Clement et al. (2014) who showed that isopycnal displacements at the RAPID western mooring locations agree well with satellite data. Specifically, Clement et al. (2014) did a similar analysis using profiles of isopycnal displacements, instead of dynamic height, from the moorings and satellite. They found statistically significant correlation (r = 0.5 - 0.6) between WB3 and WB2 with SSHA, respectively, in the upper 1000 m over 2004 to 2011. They did not analyse moorings in the eastern boundary. Given that here we have referenced the dynamic height to SLA at the surface, as well as a longer time period and different filtering, the discrepancies between our studies are reasonable.

For completeness, the SLA is also compared to dynamic height anomaly thickness for the 0-1100 dbar layer (Fig. S2). The dynamic thickness is calculated from the difference in dynamic height (referenced to 4820 dbar) at 0 and 1100 dbar and gives a measure of the shear between those levels (independent from choice of reference level). In both the western and eastern basin, the dynamic height thickness anomaly has a standard deviation smaller (0.30 $m^2$ $s^{-2}$ at WB3 and 0.07 $m^2$ $s^{-2}$ at East) than the SLA multiplied by gravitational acceleration (0.70 $m^2$ $s^{-2}$ in the west and 0.14 $m^2$ $s^{-2}$ in the east). This result suggests that the baroclinic structure in the upper 1100 dbar is not overwhelming the SLA signal, and the SLA can be used for transport estimates in the upper 1100 dbar.

[Figure]

**Figure S2.** The dynamic height thickness anomaly at moorings (a) WB3 and West, and (b) EB moorings. The dynamic heigh thickness is compared with the SLA scaled by gravitational acceleration in the western and eastern basin, respectively. Units in $m^2$ $s^{-2}$.

40. Line 180: "18-month smoothing" Is this a running mean? Is data lost at the beginning and end of your records? Please provide more details of the calculation.

Further details about the 18-month smoothing are now added in the methods section in L210-213: In this study, all 12-hour and daily data are monthly averaged, the annual seasonal climatology is removed and then smoothed with a 18-month filter over the available time periods, except where indicated otherwise. The filter is a Gaussian weighted moving average filter, where endpoints are computed such that when the number of points available is fewer than the window size, the window is resized/truncated and the average is taken from the elements available in the truncated window.

41. Lines 180-181: Please also add recent references.

A recent review paper by Hirschi et al., (2020) on the AMOC in high resolution models cites Hirschi et al (2007), Cabanes et al., (2008), and Hirschi et al., (2009) on the topic of variability in the basinwide density field, westward propagation, and AMOC transport. These references are now included in this section in L395.

42. Line 219: should be 1000 dbar?

The 100 dbar was in reference to the first peak, we have now clarified this in the text. Thank you for pointing out this typo.

43. Line 264-265: do you mean that high frequency variability becomes more important (?)

For clarification, we've added the following text in L525: Kanzow et al. (2009) suggest that near (within 100 km) of the western boundary, variability becomes influenced by a combination/mixture of barotropic-baroclinic flow over the slope, reduced eddy variability, and coastally trapped waves.

44. Line 265: Which are the limitations of the value of the slope of 0.25 (is this valid across all the array at 26.5°N or at a specific location). The accuracy in the calculation due to selecting this particular slope value should be discussed.

We can only compare altimetry with mooring values where the moorings are located at 26.5N (as we've shown in Fig.5a). There is some sensitivity in choice of altimetry location as can be seen in figure from the figure 5a (compare SLA at 26N (dashed lines) with the SLA at 28N (solid lines)). This is discussed in the text. Further discussion of the scale factor (as well as Fig. S3) is now added to this section (4.3) and rewritten for clarity (L503-528): Where $F_1(z = 0)$ is the surface value of the first mode, and $s$ is a scale factor equal to 0.25, needed to adjust $\eta$ to the correct magnitude. SLA does not account for baroclinic shear in the upper 1100 m of the water column: if we integrated the SLA over the upper 1100 m to obtain geostrophic transport, the SLA would overestimate the transport magnitude compared to dynamic height (e.g. S3). For this reason, it is necessary to combine the SLA with the vertical structure of the first baroclinic mode; however, a comparison between the dynamic height from the RAPID moorings and the SLA indicate further correction/scaling could be needed. Thus a scale factor is determined empirically by examining the signal from $\phi$ at the surface (z = 0) at moorings West and EB against $\eta$.

Fig. 5a shows the RMS dynamic height and sea level anomaly ($\eta$ multiplied by gravitational acceleration) at two latitudes across the Atlantic. At the eastern boundary, the RMS of g$\eta$ is about twice (0.18 m$^2$ s$^{-2}$) that of the dynamic height from the moorings (0.08 m$^2$ s$^{-2}$) (Fig. 5a). Moving west, RMS values of g$\eta$ peak at 74°W and 75°W (RMS of g$\eta$(27.875°N, 74°W) = 0.70 m$^2$ s$^{-2}$ and RMS of g$\eta$(26.125°N, 75°W) = 0.59 m$^2$ s$^{-2}$) before decreasing to the western boundary (RMS of g$\eta$(27.875°N, 77°W) = 0.56 m$^2$ s$^{-2}$ and RMS of g$\eta$(26.125°N, 77°W) = 0.39 m$^2$ s$^{-2}$). The rapid decrease in sea level variability at the western boundary is due to mesoscale suppression

associated with the continental slope (Kanzow et al., 2009). The moorings, in contrast, show lower variance in dynamic height at both the eastern and western boundaries (RMS of $\phi$ (26.5°N, 76.75°W) = 0.31 m² s⁻² and RMS of $\phi$ (East) = 0.08 m² s⁻²). A scatterplot of the east-west difference in surface $\phi$ (i.e. $\Delta\phi = \phi_{East}(z=0) - \phi_{West}(z=0)$) and the equivalent from sea level anomaly ($\Delta g\eta = g(\eta(27.875°N, 13.125) - \eta(27.875°N, 74.375))$) shows general agreement between the two values. The slope of the regression line is ~0.25, and the intercept goes through the origin (1.3e-17) (Fig. 5b). Fig. 5 suggests that the satellite altimetry alone does not capture the same magnitude of signal observed by the moorings. One of the reasons for this discrepancy may be due to the proximity of the moorings (e.g. WB2) to land, where variability experiences changes due to coastal processes (Kanzow et al., 2009). Kanzow et al. (2009) suggest that near (within 100 km) of the western boundary, variability becomes influenced by a combination/mixture of barotropic-baroclinic flow over the slope, reduced eddy variability, and coastally trapped waves. The value of the slope of the regression line, i.e. 0.25, is thus used as the scale factor for the satellite where indicated (e.g. Eq. 10).

[Figure]

**Figure S3.** Time averaged east-west difference in dynamic height, $\Delta\phi$ (a), and SLA multiplied by gravitational acceleration, $g\Delta\eta$ (b), at every pressure level. Units in m² s⁻².

45. Line 288: Perhaps reference Szuts et al. (2012)?

The Szuts et al. (2012) reference has now been added.

46. Line 290: add "the" between "at" and "eastern"

Done.

47. Line 312: Are these correlations only marginally significant at the 90%? It would be informative to know the threshold r for he significance fir each confidence level to facilitate

interpretation of the results. This information could be in an Appendix if it cannot be included in the main text. More interpretation of the correlations is needed.

The significance levels for Figure 9 are now revised to reflect optimal significance levels, e.g. some correlations in Figure 9 are actually statistically significant at 99% level and all are statistically significant on at least 95% level (see L665-690). Further interpretation in the context of previous studies in L655-663: The results shown here are comparable to a satellite and RAPID moorings comparison featured in Szuts et al., (2012; their Fig. 7), where correlation between SSH and geopotential anomalies from WB3 and WB2 where approximately r = 0.5, 0.7, respectively, and r = 0.75 at EB1 and close to zero at EBH. In Szuts et al., (2012) a 10-day low pass filter was used prior to correlation. Contrasted with results here, which include a longer time period, 18-month smoothing, and differences in satellite location, it is not surprising there are differences to results from Szuts et al., (2012). Szuts et al., (2012) further posited those differences between mooring and satellite are likely due to a) loss small spatial and temporal variability that occurs with satellite products, and b) satellite includes surface-layer processes which moorings do not entirely capture.

48. Line 317: Is this trend significant? Lines 316-317: How large are these trends? Are them significant?

Mention of the trend has been removed. They are not likely significant and not an integral part of the results.

49. Line 332: Please add "constructed" between "is" and "using" (second mention)

Done.

50. Line 339-34: Please add reference to the previous study for the correlation between the satellite estimate and the cable time series. I suggest adding "Similar to a previous study (Volkov et al., 2020)…."

A description/comparison of the Volkov et al., (2020) study was moved to L716. Further background information of the Gulf Stream has been added to the start of this section (5) and an additional figure showing a correlation map between the SLA and cable-based GS time series has been added to the supplementary materials (Fig. S5): The Gulf Stream within the Florida Straits has a mean of 31.2 Sv and is balanced by the UMO and Ekman transports, mean of -18 and 3.74 Sv, respectively, to yield the total mean AMOC transport of 17 Sv (McCarthy et al., 2015). The Gulf Stream time series is based on a submarine telephone cable that has been recording data between the Bahamas and Florida at 27°N since 1982 (Baringer and Larsen, 2001). Principles of geostrophy have been previously used to provide alternative mechanisms for estimating the cable-based $T_{GS}$ using satellite altimetry (Volkov et al., 2020) and pressure gauges (Meinen et al., 2020).

Here, the east-west difference in $\eta$ ($\Delta\eta$) in the western end of the basin (i.e. west of 77°W) is compared with the submarine cable data (Baringer and Larsen, 2001). Maximum correlation between $T_{GS}$ from the cable data and the satellite $\Delta\eta$ (r = 0.70, statistically significant at 95% level) is found when using $\eta_E$ located at 27.625°N and 77.125°W, and $\eta_W$ located at 27.625°N and 80.125°W (Fig. S5). Similarly, Volkov et al., (2020) used along-track satellite altimetry to infer $T_{GS}$ measured at the level of the Florida Straits, where they found that satellite altimetry captures 56% of variability observed from the $T_{GS}$ estimated from submarine cable records for the 2006-2020 time periods, where the cable-based transport estimates were subsampled at 10-day intervals to

coincide with the along-track satellite passes. We use gridded altimetry with a similar objective (to determine the Gulf Stream transport via the Florida Straits from satellite altimetry) but with a focus on the lower-frequency variability rather than an instantaneous comparison with cable measurements.

[Figure]

**Figure S5.** a) Correlation map between east-west difference in SLA ($\Delta\eta$) and the telephone cable Gulf Stream transport ($T_{GS}$). Magenta squares in the western and eastern part of basin indicate region of maximum correlation between $\Delta\eta$ and $T_{GS}$.

51. Lines 381-382: "This suggests that AMOC variability can be estimated from surface geostrophic velocity and the time-averaged vertical structure of the flow" On which time scales? Which fraction of the AMOC variability can be recovered? I suggest including this information in the abstract as well.

Thank you for this suggestion. Clarification has now been added to specify the AMOC variance captured (69%) and on what timescales (interannual) in L811-812. This information has also been included in the abstract.

52. Line 450: In general, it would be interesting to know how correlation increases/decreases for different filtering windows to provide a context for the choice of the 18-month window.

Thank you for this suggestion. Sensitivity tests were run for smoothing (at increasing 6-month intervals) of 6 month, 12-month, 18-months, and 24 months, which are now discussed throughout the text (see replies to 27, 38), as well as in section 5 (L770-789): Sensitivity of these results to the choice of filtering window was tested. We found that the correlations decreased for $T_{GS}^*$ (r = 0.6, r = 0.68) $T_{UMO}^*$ (r = 0.42, r = 0.67) when using a 6-month Gaussian filter and a 12 month Gaussian filter, respectively (Figs. S6, S7), compared to the 18-month Gaussian filter. Conversely, correlations increased for $T_{GS}^*$ (r = 0.73) $T_{UMO}^*$ (r = 0.79) when using a 24-month Gaussian filter (Fig. S8).

And in L804-807: Using a smaller filtering window, which would allow for higher frequency variability, we find the correlation for $T^*_{MOC}$ decreases (r = 0.71, r = 0.81) to for 6-month and 12-month Gaussian filters (Fig. S5, S6), respectively, but increases (r = 0.85) when using a 24 month Gaussian filter (Fig. S7).

And in the summary and conclusions section (L924-927): Sensitivity testing using different filtering windows decreases variance captured by $T^*_{UMO}$ and $T^*_{MOC}$, respectively, to 18% of $T_{UMO}$ and 50% of the $T_{MOC}$ when using 6 month Gaussian smoothing; 45% of $T_{MOC}$ and 66%of  the $T_{MOC}$ when using 12 month Gaussian smoothing; and an increase to 62% of $T_{UMO}$ and 72% of the $T_{MOC}$ when using 24 month Gaussian smoothing.

53. Line 454: Given that the time-varying vertical modes do not necessarily improve correlation between the RAPID AMOC and the satellite-derived AMOC, what does this mean for the new dynamical method proposed? It is not clear if this is an improvement from the Frajka-Williams (2015) study or not. If I understand correctly, their satellite-based estimate recovers over 90% of the interannual variability of the MOC measured by the RAPID 26°N array, but if I interpret correctly with this new method developed in this study a smaller fraction of the variance is recovered from the satellite based estimate. The Summary and conclusions section misses interpretation and does not give any conclusion about the usefulness (or not) of this new method and for which time scales of variability, and whether the recommendation is to apply it or not.

Thank you for this suggestion. We have now added clarification that has hopefully improved the interpretation in L899: One previous effort to use satellite altimetry as a proxy for AMOC variability, led by EFW15, relied on a statistical relationship between the RAPID $T_{UMO}$ and the SLA on low frequency (18 month) timescales to estimate the  $T_{UMO}$. The EFW15 $T_{MOC}$ was estimated by adding the same estimates of $T_{EK}$ (ERA5 reanalysis) and $T_{GS}$ (submarine cable) used by RAPID to the $T_{UMO}$. Their method successfully recovered ~85% of the $T_{UMO}$ and 92% of the $T_{MOC}$ on interannual time scales. Earlier studies used more dynamically based methods and combined SLA with the vertical structure of horizontal flow from mooring data and found that the SLA had little skill at capturing the upper mid-ocean transport (e.g. Hirschi et al., 2009; Kanzow et al., 2009; Szuts et al., 2012). The goal of this study was to similarly use satellite altimetry as a proxy for AMOC variability, and re-evaluate dynamics-based method (e.g. Hirschi et al., 2009; Kanzow et al., 2009; Szuts et al., 2012) for using satellite altimetry to estimate the $T_{UMO}$ as well as the $T_{GS}$ components of the AMOC, which could eventually be used at other latitudes. Thus, using principles of geostrophy and normal mode decomposition a method for constructing the upper mid-ocean transport, $T^*_{UMO}$, and by extension the $T^*_{MOC}$, at 26°N from satellite altimetry on low frequency (18 months) time scales was devised here by combining the first baroclinic mode, derived from time-averaged density profiles from the RAPID moorings, with SLA to reproduce the $T^*_{UMO}$ and $T^*_{MOC}$ transports. Using this new method, we find that 56% of RAPID $T_{UMO}$ variability could be captured and 69% of the RAPID $T_{MOC}$ variability on interannual (18 month) time scales. Sensitivity testing using different filtering windows decreases variance captured by $T^*_{UMO}$ and $T^*_{MOC}$, respectively, to 18% of $T_{UMO}$ and 50% of the $T_{MOC}$ when using 6 month Gaussian smoothing; 45% of $T_{MOC}$ and 66%of  the $T_{MOC}$ when using 12 month Gaussian smoothing; and an increase to 62% of $T_{UMO}$ and 72% of the $T_{MOC}$ when using 24 month Gaussian smoothing. The $T_{GS}$ component of $T_{MOC}$ was also reproduced separately using the satellite altimetry; we found that it captures 49% of the GS variability via the Florida Straits as measured from the telephone cable (Meinen et al., 2010) on interannual time scales, similar to a recent study by Volkov et al. (2020), who also used satellite to estimate GS variability in the Florida Straits. Finally, the satellite-derived $T^*_{UMO}$ and $T^*_{MOC}$ are reconstructed for the full satellite period (1993 to 2018). The 26-year

reconstructions of $T^*_{UMO}$ and $T^*_{MOC}$ were compared with another separate satellite-derived proxy (EFW15) and a $T_{MOC}$ reconstruction from the GloSea5 reanalysis. Though all reconstructions had correlations of at least r = 0.79 with RAPID-based transports for the 2004 to 2013 time period, the 1993-2003 time period showed poor agreement between the three $T_{MOC}$ reconstructions, suggesting further studies are needed to confidently reproduce the fluctuations in the AMOC during times pre-dating the RAPID programme.

Because the altimetry has no knowledge of vertical shear, estimating transport over the upper 1000 m using only satellite altimetry results in an overestimation of the transport's magnitude (Fig. S2). Thus, normal mode decomposition was investigated using the RAPID moorings to answer 2 questions: a) How much does the vertical structure of the flow contribute to the upper-mid ocean transport ($T_{UMO}$) variability, and b) can the pressure modes be combined with the satellite to provide an improved way of estimating mass overturning transport. In the first instance, we find that the first baroclinic mode accounts for 83% of the observed interior geostrophic transport variability, the barotropic mode accounts for 82% of the variability, and the combined barotropic and first baroclinic mode account for 98% of the total variability. In the second instance, we find that combining the satellite altimetry with the vertical structure (from the 1st baroclinic mode) improves the magnitude of values for the altimetry-based transport. However, a scale factor is still needed to further correct the values to capture the magnitude of the $T_{UMO}$. We posit the need for a scale factor is the result of the satellite altimetry not capturing the full signal observed by the moorings, because of proximity of moorings to land, where variability is also influenced by coastal processes (Kanzow et al., 2009). It is also important to note that combining the altimetry with the vertical structure (from 1st baroclinic mode) does not improve the amount of variance captured by the satellite-based $T^*_{UMO}$ compared to RAPID $T_{UMO}$ variability, and using time-varying pressure modes (instead of time-averaged) do not necessarily improve correlation between the RAPID $T_{MOC}$ and the satellite-derived $T^*_{MOC}$. Further discrepancies between satellite-based transport and RAPID transport may be due to the barotropic component, which the satellite-based method does not account for.

To summarise, principles of geostrophy can be used with satellite altimetry to effectively capture upper mid-ocean and GS transport, and by extension, AMOC transport. The vertical structure of the flow (from the moorings) does not improve the amount of variability the satellite-based method captures; however, the analysis of the normal mode decomposition yields insight into the governing modes of variability associated with the $T_{UMO}$ on interannual timescales, and the limits this places on any satellite-based transport method. While this method progresses attempts at a dynamically robust method for estimating AMOC transport using satellite altimetry, which can also be used to recreate the AMOC the start of the satellite time period, pre-dating RAPID, the method is still not independent from the mooring array and without improved understanding of the scale factor, further investigation is needed to produce an altimetry-based method for the AMOC that can be used at other latitudes.

54. I miss a discussion on how these new estimates from the satellite-based dynamical method accounting for the full-depth vertical structure of the flow at the mooring sites would be validated should there be no moorings to provide this information. Would the first BC mode and BT mode recovered from the data since 2004 enough?

Thank you. Yes, results here show as state in section 5 (L809-811) and in conclusions (L969) that the time-averaged vertical structure is enough for transport estimates on interannual time scales, so modes recovered from current time series would be enough.

55. I suggest the authors discussing the advantages/disadvantages of this new dynamically-based method in comparison with the previous published work and providing a recommendation about which method they propose for substitute for AMOC estimates at 26.5°N and on which time scales.

New text to this effect has been added in L974-982: To summarise, principles of geostrophy can be used with satellite altimetry to effectively capture upper mid-ocean and GS transport, and by extension, AMOC transport. The vertical structure of the flow (from the moorings) does not improve the amount of variability the satellite-based method captures; however, the analysis of the normal mode decomposition yields insight into the governing modes of variability associated with the $T_{UMO}$ on interannual timescales, and the limits this places on any satellite-based transport method. While this method progresses attempts at a dynamically robust method for estimating AMOC transport using satellite altimetry, which can also be used to recreate the AMOC the start of the satellite time period, pre-dating RAPID, the method is still not independent from the mooring array and without improved understanding of the scale factor, further investigation is needed to produce an altimetry-based method for the AMOC that can be used at other latitudes.

56. Line 462: This is perhaps a matter of taste, but I suggest replacing "monitoring" for "estimates".

Done.

57. Figure 2: The black square at the EB is hard to visualize. I suggest either zooming in for this region or choosing another color for the squares (magenta?). The mooring positions are very hard to see. I suggest using smaller symbols so that the moorings symbols do not appear on top of each other. I suggest labeling West, WB3 and EB on the map to guide the reader (specially fro WB3 position).

Thank you for your suggestions. The squares are now in magenta with black outline. To make the mooring positions easier to see an inset with their location and labels is now added to Figure 1.

58. Figures 1, 2: The moorings at the shallowest region near the EB are very difficult to visualize

An inset with clearer indication of mooring positions and labels is now included in Figure 1. See also reply to comment 57.

59. Figure 3: Please add information in the figure caption about the SLA product used or refer to the section in the paper where that information can be found. The colorbar is not clear. Is the last level contoured from 0.13 to 0.14 m? (if so the last level should be labeled on the colorbar) or the last level refers to values above 0.13 m? Please revise. The a), b) labels for the left and right panels are difficult to visualize.

Thank you for your suggestions. Further information has now been added about the SLA product, the colorbar now has more labels, and the a), b) labels have now been move outside of the panels for clarity.

References (not exhaustive)

Cunningham, S. A., T. Kanzow, D. Rayner, M. O. Baringer, W. E. Johns, J. Marotzke, H. R. Longworth, E. M. Grant, J. J-M. Hirschi, L. M. Beal, C. S. Meinen, and H. L. Bryden, "Temporal Variability of the Atlantic Meridional Overturning Circulation at 26.5°N", Science, 317, 935, DOI: 10.1126/science.1141304, 2007.

Herrford, J., Brandt, P., Kanzow, T., Hummels, R., Araujo, M., & Durgadoo, J. V. (2021). Seasonal variability of the Atlantic Meridional Overturning Circulation at 11° S inferred from bottom pressure measurements, Ocean Sci., 17, 265–284, https://doi.org/10.5194/os-17-265-2021, 2021.

Hummels, R., Brandt, P., Dengler, M., Fischer, J., Araujo, M., Veleda, D., & Durgadoo, J. V. (2015). Interannual to decadal changes in the western boundary circulation in the Atlantic at 11°S. Geophysical Research Letters, 42, 7615– 7622, doi:10.1002/2015GL065254.

Johns, W. E., M. O. Baringer, L. M. Beal, S. A. Cunningham, T. Kanzow, H. L. Bryden, J. J.-M. Hirschi, J. Marotzke, C. S. Meinen, B. Shaw, and R. Curry, Continuous Array-Based Estimates of Atlantic Ocean Heat Transport at 26.5°N, J. Clim., 24(5), 2429-2449, doi:10.1175/2010JCLI3997.1, 2011.

Kanzow, T., S. A. Cunningham, W. E. Johns, J. J-M. Hirschi, J. Marotzke, M. O. Baringer, C. S. Meinen: Seasonal variability of the Atlantic meridional overturning circulation at 26.5°N, J. Clim., 23(21), 5678-5698, 2010.

Kanzow, T., S. A. Cunningham, D. Rayner, J. J-M. Hirshi, W. E. Johns, M. O. Baringer, H. L. Bryden, L. M. Beal, C. S. Meinen, and J. Marotzke, "Observed flow compensation associated with the Meridional Overturning at 26.5°N in the Atlantic", Science, 317, 938, DOI: 10.1126/science.1141293, 2007.

Kersalé, M., Meinen, C. S., Perez, R. C., Le Hénaff, M., Valla, D., Lamont, T., et al. (2020). Highly Variable Upper and Abyssal Overturning Cells in the South Atlantic. Science Advances, Vol. 6, no. 32, eaba7573, doi: 10.1126/sciadv.aba7573, 2020.

Lozier, M.S., F. Li, S. Bacon, F. Bahr, A. Bower, S. Cunningham, F. de Jong et al.: A sea change in our view of overturning in the Subpolar North Atlantic Program. Science 01 Feb 2019:Vol. 363, Issue 6426, pp. 516-521 DOI: 10.1126/science.aau6592

Meinen, C. S., M. O. Baringer, and R. F. Garcia, "Florida Current Transport Variability: An Analysis of Annual and Longer-Period Signals", Deep Sea Res. I, 57 (7), 835-846, doi:10.1016/j.dsr.2010.04.001, 2010.

Meinen, C. S.; Speich, S.; Piola, A. R.; Ansorge, I.; Campos, E.; Kersalé, M. et al..: Meridional Overturning Circulation Transport Variability at 34.5∘S during 2009-2017: Baroclinic and Barotropic Flows and the Dueling Influence of the Boundaries, Geophys. Res. Lett.;10.1029/2018GL077408, 2018.

**Reviewer #2**

Review of Ocean Sciences 2021-10

This article presents an estimate of the overturning transport at 26N based solely on satellite data. In doing so, it investigates the structure and correlation of each part of the calculation, with an emphasis on the mid-ocean transport.

This is a worthwhile publication that combines previous findings with new analyses and creates self-consistent analysis that will be of use to the field. It is well written and has graphics that are a pleasure to look at. My main concern is that there are a number of steps that lack a physical motivation and seem ad hoc, especially so because the purpose of the approach taken is not made explicit. More generally, the paper would be clearer if it were more explicit about its goals, what is novel, and how how it compares with earlier studies.

We thank the reviewer for their constructive comments.

We have now revised the manuscript to include further background information and discussion from previous studies to give better context to results from this work. We have also re-written some sections to add clarity on the novelty of this work and its advantages and disadvantages compared to previously published methods.

A supplementary materials section has now been added with new analysis showing the results of sensitivity testing depending on the choice of Gaussian filtering of the data on the results (Fig. S1, S6, S7, S8). An additional figure comparing SLA to dynamic heigh anomaly thickness has been added to provide a more complete discussion of the relationship between SLA and dynamic height (Fig. S2) in section 3. Further discussion of the scale factor a new figure showing east-west difference in dynamic height vs SLA (Fig. S3) has been added to section 4.3. A new figure showing transport anomalies from the reconstructed dynamic height anomaly for the first baroclinic mode, the second baroclinic mode, and the barotropic mode has been added and discussed in section 4.4 (Fig. S4). Finally, a new figure showing the correlation map between the SLA in the Florida Straits and the cable-based Gulf Stream transport (Fig. S5) is added and discussed in section 5.

Details for all these changes are described in the reply to specific comments below. All reviewer comments are itemized and addressed in blue. Line numbers given here are in reference to the tracked changes manuscript. New text added to the manuscript is in red.

Detailed comments:

1. Abstract: I did not get a clear sense of how this paper fits into the existing literature from reading this abstract, and what analyses were done and to what immediate purpose.

Details on relevant background literature and the associated knowledge gap have now been significantly expanded on in the entire introduction section.

2. Equation 4: Since surface geostrophic velocity is not treated further, suggest removing this equation.

Equation 4 is now removed.

3.  section 3.2, figure 2b.  This section and the data presented are not presented in a clear manner.   It is not correct to correlate two quantities that have a common signal (SLA) and interpret statistically significant correlation as "the variability at the sea surface is a good measure and coherent with variability to at least 1000 dbar".  The reader could readily counter by stating that the correlation is meaningless, as correlating A (=SLA) and A+B (=SLA+dynamic height) just shows that A is coherent with A.  Either correlate A=SLA and B=dyn height and discuss those results, or discuss that SLA provides greater than 64% of the variance of the surface-reference dynamic height, and thus that SLA is more important for transport than vertical structure of dynamic height

The point of figure 2b is to illustrate how well and to what depth SLA captures sub-surface variability. As suggested by the reviewer, we have also added a new plot in the supplementary materials (Fig. S2) showing the dynamic height anomaly thickness (0 -1100 dbar) which provides a measure of shear independent of the reference level. Corresponding text is added in L378-384: For completeness, the SLA is also compared to dynamic height anomaly thickness for the 0-1100 dbar layer (Fig. S2). The dynamic thickness is calculated from the difference in dynamic height (referenced to 4820 dbar) at 0 and 1100 dbar and gives a measure of the shear between those levels (independent from choice of reference level). In both the western and eastern basin, the dynamic height thickness anomaly has a standard deviation smaller (0.30 $m^2$ $s^{-2}$ at WB3 and 0.07 $m^2$ $s^{-2}$ at East) than the SLA multiplied by gravitational acceleration (0.70 $m^2$ $s^{-2}$ in the west and 0.14 $m^2$ $s^{-2}$ in the east). This result suggests that the baroclinic structure in the upper 1100 dbar is not overwhelming the SLA signal, and the SLA can be used for transport estimates in the upper 1100 dbar.

[Figure]

**Figure S2.** The dynamic height thickness anomaly at moorings (a) WB3 and West, and (b) EB moorings. The dynamic heigh thickness is compared with the SLA scaled by gravitational acceleration in the western and eastern basin, respectively. Units in $m^2$ $s^{-2}$.

4. Equation (5) is incomplete because it doesn't reflect that SLA is added, as described in the text.

Equation 5 shows how dynamic height is estimated from the moorings and was moved to the methods section (section 2.1, L182), to clarify how dynamic height is computed, where it is more appropriately placed. It is now equation 1 and used before the idea of adding SLA was introduced.

5. Figure 2b. The two-tailed t-value confidence limits on correlation seem inappropriate for this case: is the confidence limit of correlating SLA with itself (at p=0) truly non-zero? Maybe this reflects that it is hard to get physical meaning out of correlating A with A+B, and that the statistical question or the conclusion needs to be posed a different way.

Please see reply to comment 3.

6. line 219. It is not physically possible for the buoyancy frequency to be zero, unless the water is perfectly homogeneous (which it isn't). Suggest plotting N(z) in Fig 4 in log-space to more clearly show its structure.

Apologies, this was indeed an error in interpretation based on the linear N(z) plot. N(z) is now plotted in log-space on the X axis, and the comment in the text has been updated (L447: "before decreasing to background stratification levels around 1100 dbar").

[Figure]

7. lines 250-265. This is a very dense paragraph with lots of numbers and shifting references that was difficult to comprehend.

Thank you for pointing this out. That paragraph (now L511-528) has been rewritten for clarity: Fig. 5a shows the RMS dynamic height and sea level anomaly ($\eta$ multiplied by gravitational acceleration) at two latitudes across the Atlantic. At the eastern boundary, the RMS of $g\eta$ is about twice (0.18 $m^2$ $s^{-2}$) that of the dynamic height from the moorings (0.08 $m^2$ $s^{-2}$) (Fig. 5a). Moving west, RMS values of $g\eta$ peak at 74°W and 75°W (RMS of $g\eta$(27.875°N, 74°W) = 0.70 $m^2$ $s^{-2}$ and RMS of $g\eta$(26.125°N, 75°W) = 0.59 $m^2$ $s^{-2}$) before decreasing to the western boundary (RMS of $g\eta$(27.875°N, 77°W) = 0.56 $m^2$ $s^{-2}$ and RMS of $g\eta$(26.125°N, 77°W) = 0.39 $m^2$ $s^{-2}$). The rapid decrease in sea level variability at the western boundary is due to mesoscale suppression associated with the continental slope (Kanzow et al., 2009). The moorings, in contrast, show lower variance in dynamic height at both the eastern and western boundaries (RMS of $\phi$ (26.5°N, 76.75°W) = 0.31 $m^2$ $s^{-2}$ and RMS of $\phi$ (East) = 0.08 $m^2$ $s^{-2}$). A scatterplot of the east-west difference in surface $\phi$ (i.e. $\Delta\phi = \phi_{East}(z = 0) - \phi_{West}(z = 0)$) and the equivalent from sea level anomaly ($\Delta g\eta = g(\eta(27.875°N, 13.125) - \eta(27.875°N, 74.375))$) shows general agreement between the two values. The slope of the regression line is ~0.25, and the intercept goes through the origin (1.3e-17) (Fig. 5b). Fig. 5 suggests that the satellite altimetry alone does not capture the same magnitude of signal observed by the moorings. One of the reasons for this discrepancy may be due to the proximity of the moorings (e.g. WB2) to land, where variability experiences changes due to coastal processes (Kanzow et al., 2009). Kanzow et al. (2009) find that the signal is reduced near (within 100 km) to the western boundary, thus altimetry may overestimate variability in this region. The value of the slope of the regression line, i.e. 0.25, is thus used as the scale factor for the satellite where indicated (e.g. Eq. 10).

8. equation 11. What is the physical reason for adding a scale factor to this equation? What purpose does it serve? This seems like an ad hoc decision, and it needs to be justified. How the scale factor is computed also needs to be described clearly. The sentence "The scale factor was determined ... against eta." (lines 250-251) is insufficient, and further contradicts the description later in the same paragraph that the scale factor comes from fitting _differences_ of eta (shown in fig 5b). It is not logical that a scale factor for eta is related to a scale factor for differences of eta.

Further justification and an additional figure (S3) has now been added to the text (L503-509): Where $F_1(z = 0)$ is the surface value of the first mode, and $s$ is a scale factor equal to 0.25, needed to adjust $\eta$ to the correct magnitude. SLA does not account for baroclinic shear in the upper 1100 m of the water column: if we integrated the SLA over the upper 1100 m to obtain geostrophic transport, the SLA would overestimate the transport magnitude compared to dynamic height (e.g. S3). For this reason, it is necessary to combine the SLA with the vertical structure of the first baroclinic mode; however, a comparison between the dynamic height from the RAPID moorings and the SLA indicate further correction/scaling could be needed. Thus a scale factor is determined empirically by examining the signal from $\phi$ at the surface (z = 0) at moorings West and EB against $\eta$.

[Figure]

**Figure S3.** Time averaged east-west difference in dynamic height, $\Delta\phi$ (a), and SLA multiplied by gravitational acceleration, $g\Delta\eta$ (b), at every pressure level. Units in $m^2\ s^{-2}$.

Also please see reply to comment 7.

9. Please provide confidence limits for the slope shown in 5b, providing a correlation coefficient R would be nice too. To my eye, this plot seems to show a deficiency of least squares in underestimating the slope because there is assumed to be no error in the dependent values. A principal component analysis or alternate least squares formulations are needed to account for uncertainty in both dependent and independent variables.

Confidence limits for the slope now included in the Fig. 5b.

10. lines 263-265 "The comparison between phi(z=0) ... observed by the moorings". This logic doesn't make sense to me. To play devil's advocate, if the altimetric SLA has a higher signal than the moorings (fig 5a), then shouldn't the interpretation be that the SLA (being larger) captures more of the signal than the mooring? There's lots more going on, of course.

Thank you for pointing this out, we have now clarified in the text to say "satellite altimetry alone does not capture the same magnitude of signal observed by the moorings" and the comment on the sources of 'missing' variability in L525: Kanzow et al. (2009) suggest that near (within 100 km) of the western boundary, variability becomes influenced by a combination/mixture of barotropic-baroclinic flow over the slope, reduced eddy variability, and coastally trapped waves.

11. Fig 5a. This figure shows dynamic height, but, because dynamic height is less than the rms of SLA, it seems like dynamic height is _not_ referenced to SLA. In contrast, previous discussion (section 3.2, fig 2b) clearly do reference dynamic height to SLA. Please make

clearer what quantity is being used, and preferably use the same quantity consistently throughout the paper. If there's merit to use dynamic height referenced to SLA in some cases, and straight dynamic height in other cases, then please add reasoning to explain what insight is provided by using both methods.

Thank you for pointing this out, we have now clarified in L348 that dynamic height is referenced to SLA only in section 3.2 and Fig. 2b, and have assigned a different symbol for dynamic height when referenced to SLA at the surface: $\phi_\eta$.

12. line 274-275 For the trend in mode values to be real, as stated here, requires that the mode fits are completely stationary over the 15 years of records. Two factors need to be investigated before this conclusion can be reached. First, does the stratification changes over the time series? Using a constant stratification for the mode fits assumes stationarity. Second, do the sampling depths on the moorings remain constant from 2004 to 2018? Changes in sampling depths can easily change how the CTD profiles project onto vertical modes.

Mention of the trend has been removed. They are not likely significant and not an integral part of the results.

13. lines 317-318 "All three transport estimates show slowly increasing trend over the 13 year period at West and EB". Trends like this can also result from the comment above, about how either sampling depths or stratification is not stationary over this 13 year span.

Thank you for these suggestions. Mention of the trend has been removed. They are not likely significant and not an integral part of the results.

14. line 323, fig 9f. It is practically impossible for 2 independent time-series to have a correlation of 1. Please provide more significant figures for this R value instead of rounding it up.

Figure 9 now has 3 digits of precision for the r values.

15. line 323-325. Instead of saying the barotropic mode "plays a non-negligible role in the total variance", why not quantify its variance, as can easily be done with the numbers presented here?

Thank you for this suggestion. The contribution of the barotropic mode is now quantified in the abstract, results, and conclusions. There is also a new figure (S5) quantifying the contribution from the barotropic and the second baroclinic mode separately (see section 4.4) in L669: Analysis of the barotropic component shows that it accounts for 86% and 63%, respectively, in the total variability at West and EB (Fig. S4) and could explain some of the discrepancies in the satellite-based estimates, as satellite altimetry only captures baroclinic variability. The second baroclinic mode is shown to account for 18% and 0.34% (not statistically significant) of variance at West and EB moorings (Fig. S4).

And in L701-704: These results suggest that the time-averaged first baroclinic mode accounts for most of the interior geostrophic transport variability; while the barotropic mode accounts for 82% of the variability, and the combined barotropic and first baroclinic mode accounts for 98% (Fig. S4). The barotropic mode is reflective of changes in the deeper less stratified ocean.

16. Section 5. There have been studies done by people at AOML about using sea level to estimate the Gulf Stream Transport, and perhaps even using SSH, that would be good to reference in this section. Many details are skipped over here - such as the SLA difference going across the Bahamas, the mismatch of time-scales between the cable voltage measurements, the SLA values, and the satellite altimeter results of Volkov et al. If these details are not important for this section, then say what the goal is succinctly - to identify the most accurate altimetry-based proxy for T_GS?

Thank you for this suggestion. Further background information of the Gulf Stream has been added to the start of this section (5) and an additional figure showing a correlation map between the SLA and cable-based GS time series has been added to the supplementary materials (Fig. S4): The Gulf Stream within the Florida Straits has a mean of 31.2 Sv and is balanced by the UMO and Ekman transports, mean of -18 and 3.74 Sv, respectively, to yield the total mean AMOC transport of 17 Sv (McCarthy et al., 2015). The Gulf Stream time series is based on a submarine telephone cable that has been recording data the Bahamas and Florida at 27°N since 1982 (Baringer and Larsen, 2001). Principles of geostrophy have been previously used to provide alternative mechanisms for estimating the cable-based $T_{GS}$ using satellite altimetry (Volkov et al., 2020) and pressure gauges (Meinen et al., 2020).

Here, the east-west difference in $\eta$ ($\Delta\eta$) in the western end of the basin (i.e. west of 77°W) is compared with the submarine cable data (Baringer and Larsen, 2001). Maximum correlation between $T_{GS}$ from the cable data and the satellite $\Delta\eta$ (r = 0.70, statistically significant at 95% level) is found when using $\eta_E$ located at 27.625°N and 77.125°W, and $\eta_W$ located at 27.625°N and 80.125°W (Fig. S4). Similarly, Volkov et al., (2020) used along-track satellite altimetry to infer $T_{GS}$ measured at the level of the Florida Straits, where they found that satellite altimetry captures 56% of variability observed from the $T_{GS}$ estimated from submarine cable records for the 2006-2020 time periods, where the cable-based transport estimates were subsampled at 10-day intervals to coincide with the along-track satellite passes. We use gridded altimetry with a similar objective (to determine the Gulf Stream transport via the Florida Straits from satellite altimetry) but with a focus on the lower-frequency variability rather than an instantaneous comparison with cable measurements.

[Figure]

**Figure S4.** a) Correlation map between east-west difference in SLA (Δη) and the telephone cable Gulf Stream transport ($T_{GS}$). Magenta squares in the western and eastern part of basin indicate region of maximum correlation between Δη and $T_{GS}$.

17. equation 18. This does not make sense, the units on either side of the equation are inconsistent. What is the "8"? How is it calculated? What is its uncertainty? In any case, what is the physical reason for adding a scale factor into this equation?

The satellite-based Gulf Stream transport was estimated from linear regression using the cable-based Gulf Stream transport and the Δη where correlation was maximum (Fig. S4). The regression coefficients were then applied to Δη to produce a final estimate for the satellite-based Gulf Stream transport. This has now been clarified in the text L739: Here, linear regression is used against the cable-based $T_{GS}$ to obtain the regression coefficients (*a, b*) needed to produce a final estimate for satellite-based GS transport:

$$T_{GS}^*(t) = a\Delta\eta + b, \tag{17}$$

Where a = 8.0087, b = 0.0026, the eastern and western SLA points that correspond to Δη are located at 27.625°N and 77.125°W for $\eta_E$, and at 27.625°N and 80.125°W for $\eta_W$. The accuracy of $T_{GS}^*$ was determined using a Monte-Carlo technique, where 90% of the time series was randomly sampled 10,000 times, performing a linear regression to obtain an upper (mean value of 0.0614 Sv) and lower bound (mean value of -0.0614 Sv) for the 90% confidence intervals. The satellite-derived $T_{GS}^*$ captures 49% of the variability of the in-situ $T_{GS}$ and generally underestimates the amplitude of variability (Fig. 10a,b).

We have looked at the sensitivity of the GS transport to the regression by using a Monte Carlo method which used the confidence intervals from the regression (L745). Figure 10 shows a measure of spread (blue shading) which indicates the GS transport is not very sensitive to the regression we have used.

18. line 345.  Please provide the upper and lower bound referenced in this line.

The upper and lower bound mean values are now given.

19. lines 364-367.  How does the Monte Carlo method give an error that varies with time (as plotted in fig 10)?

The mean upper and lower bound values estimated from the Monte-Carlo technique (L767-770) are multiplied by the time series in question.

20. line 383.  Since neither "geostrophic velocity" nor "vertical structure of flow" was presented, suggest replacing with "SLA" and "dynamic height".

We have now made this change.

21. line 396.  "... to provide a more physically robust method."  This is debatable, especially given the seemingly ad hoc decisions made earlier involving scale factors.  An important advance I see is that it gives a methodologically consistent (satellite only) method that would be straight forward to apply to other latitudes.

We have now modified this sentence as suggested (L832): Here it is fitting to compare the EFW15 method with the method derived in sections 4 and 5, which relies on altimetric data and geostrophic balance, and therefore gives a methodologically consistent (satellite only) method that incorporates climatological stratification and has the potential to be applied to other latitudes. This is in contrast with EFW15 which was a statistically based method that assumes the existence of in situ observations. Further work is needed to determine the origins of the scale factor required to balance the transports, whether due to sampling issues or uncertainties in satellite sea level anomaly near the boundaries.

22. lines 423-426, 429-430.  When I look at the plotted data, I do not see the conclusions as stated in the text.

Thank you for pointing this out. "All three TMOC reconstructions show a weakening in the southward flow in 1996 or 1997, followed by an intensification toward the late 90s. This general mid-to-late 90s strengthening is in agreement with changes in the Atlantic Multidecadal Variability phase, which was marked by a positive phase in the late 1990s (Zhang et al., 2019). " has now been corrected to "The EFW15 $T_{MOC}$ reconstruction shows a weakening in the southward flow in ~1996-1997 (positive anomaly), while the $T_{MOC}^*$ shows strengthening in the southward flow in 1997(negative anomaly). None of the reconstructions quite agree with the general mid-to-late 90s strengthening in the AMOC, associated with changes in the Atlantic Multidecadal Variability phase, marked by a positive phase in the late 1990s, discussed by Zhang et al. (2019)."

23. lines 431-433.  The discussion could be advanced by mentioning ways in which these 3 data sets are non stationary.  The poor agreement between the 3 methods in the first half of the altimetry record reminds me of the problem of overfitting.  If a training data set is used to fit a model, such as done over the years of mooring measurements, then if that model is fit too closely to the data then in will not be very predictive when applied to data outside of the training set (that is 1993-2004).  The ad-hoc scaling factors used to reach this point are consistent with overfitting.

Thank you for this suggestion. We have now added discussion to this effect in L884-888: Reasons for this disagreement may be due to overfitting. In the case of the satellite-based transport

reconstruction, EFW $T_{MOC}$, the statistical method used to estimate transport relied on RAPID mooring data and resulting time series in the period pre-dating RAPID, 1993-2003, may be biased by overfitting of data during the RAPID period, 2004-2013. Similarly, $T_{MOC}^*$ may be biased by choice of the scale factor, here based on comparisons between satellite and moorings during the RAPID time period.

24. summary point, lines 445-47: Why was is necessary to add a scale factor to SLA, and what is its physical purpose?

New text has been added to conclusions and summary section (as well as section 4.3 – see reply to comment 8) to clarify the point of the scale factor L938: Because the altimetry has no knowledge of vertical shear, estimating transport over the upper 1000 m using only satellite altimetry results in an overestimation of the transport's magnitude (Fig. S3). Thus, normal mode decomposition was investigated using the RAPID moorings to answer 2 questions: a) How much does the vertical structure of the flow contribute to the upper-mid ocean transport ($T_{UMO}$) variability, and b) can the pressure modes be combined with the satellite to provide an improved way of estimating mass overturning transport. In the first instance, we find that the first baroclinic mode accounts for 83% of the observed interior geostrophic transport variability, the barotropic mode accounts for 82% of the variability, and the combined barotropic and first baroclinic mode account for 98% of the total variability. In the second instance, we find that combining the satellite altimetry with the vertical structure (from the 1st baroclinic mode) improves the magnitude of values for the altimetry-based transport. However, a scale factor is still needed to further correct the values to capture the magnitude of the $T_{UMO}$. We posit the need for a scale factor is the result of the satellite altimetry not capturing the full signal observed by the moorings, because of proximity of moorings to land, where variability is also influenced by coastal processes (Kanzow et al., 2009). It is also important to note that combining the altimetry with the vertical structure (from 1st baroclinic mode) does not improve the amount of variance captured by the satellite-based $T_{UMO}^*$ compared to RAPID $T_{UMO}$ variability, and using time-varying pressure modes (instead of time-averaged) do not necessarily improve correlation between the RAPID $T_{MOC}$ and the satellite-derived $T_{MOC}^*$. Further discrepancies between satellite-based transport and RAPID transport may be due to the barotropic component, which the satellite-based method does not account for.

25. summary point, line 450: There was no use/discussion of Rossby Wave theory. Normal mode decomposition, yes, but that's different.

This is now corrected in text.